# Transformers as Optimal Transport: Stability, Geometry, and Gauge Symmetry

## Abstract

Self-attention is row-wise entropic optimal transport: masked softmax exactly solves independent OT problems on each query's support with unit entropic regularization ($\varepsilon = 1$)—not an approximation, but a precise mathematical equivalence. This yields a compositional stability theory via a global $\ell_\infty \to \ell_1$ Lipschitz bound across heads, residuals, and LayerNorm, producing a conservative drift budget and explaining representation locking through local saturation when $\delta(P) \to 0$. We introduce gauge-invariant coarse Ricci curvature with $\tau$-dependent bounds linking temperature and key scale to contraction, and show depth behaves as Wasserstein gradient flow via an evolution variational inequality. Empirically on GPT-2 variants, measured drift sits well below theoretical budgets (tightness ratio $\approx 0.043$), locking occurs in $\sim 10\%$ of samples ($\mathrm{TV} < 10^{-10}$), Sinkhorn $W_2$ concentrates in mid-depth, and curvature gaps tighten with larger $\tau$ or smaller key scale as predicted. We prove depth cannot collapse: compositions generically lack single-layer representations with the same key dimension. We report extrinsic Euclidean quantities in a declared canonical gauge. The framework provides actionable design principles for temperature, key scaling, and early exit while organizing attention into a coherent geometric structure.

## 1 Introduction

Self-attention is central to modern sequence models, yet its stability properties and the empirical phenomenon often called *representation locking* remain only partially explained at a mechanistic level. We adopt a variational view: masked attention is exactly a collection of independent, row-wise entropic optimal transport (OT) problems on each query's masked support. Specifically, we prove that standard attention with temperature $\tau$ solves these OT problems with unit entropic regularization ($\varepsilon = 1$)—not an approximation or analogy, but a precise mathematical equivalence where $\varepsilon = 1$ emerges from the softmax structure itself. This lens yields (i) a compact, compositional Lipschitz-to-drift bound that explains stability across residual blocks and LayerNorm with measured tightness ratios of $0.043 \pm 0.021$ and (ii) a local saturation regime that quantitatively accounts for locking in $\sim 10\%$ of samples. We further add geometry via a gauge-invariant coarse Ricci curvature that captures contraction, and we show that depth behaves like a Wasserstein gradient flow up to controlled parameter drift. Finally, we formalize a head-level gauge symmetry (including the RoPE commutant) that leaves logits and masks invariant, clarifying which diagnostics are intrinsic and when a canonical gauge is required for extrinsic norms. Core statements appear in Sections Sections 2 to 5, with OT/KKT and temperature proofs in Sections A and B, softmax bounds and saturation in Section D, component-wise budgets in Section E, geometry derivations in Section F, and gauge proofs and experimental protocols in Sections I and J.

**Contributions.**

- **Exact OT with $\varepsilon = 1$.** Standard masked attention exactly solves the row-wise entropic OT program at $\varepsilon = 1$; temperature $\tau$ only rescales the cost (Section 2; Sections A and B). This makes the OT structure intrinsic to standard implementations rather than a design choice.

- **Quantitative stability and locking.** A 1-Lipschitz softmax bound in $(\ell_\infty, \ell_1)$ yields a compact per-layer drift budget that composes across heads, residual connections, LayerNorm, and probes (Section 3; Sections D and E). Local and global saturation explain vanishing updates as $\delta(P) \to 0$, and a minimal rank obstruction shows depth cannot in general be collapsed to a single layer with the same key dimension (Appendix D).
- **Geometry and depth-as-flow.** We define a coarse Ricci curvature with a lower bound linking curvature to $\tau$ and key scale, and derive an evolution variational inequality (EVI) indicating movement toward Gibbs equilibria up to drift (Section 4; Section F). Empirically, depth scaling is sublinear.
- **Gauge symmetry and canonical reporting.** A head-level gauge action on $(Q, K, V)$ leaves logits, masks, and the row-wise OT objectives/minimizers invariant; we extend to multi-head and RoPE via the commutant restriction (Section 5; Sections I and I.1). This separates intrinsic diagnostics from those requiring a declared canonical gauge.
- **Empirical validation.** Four diagnostics align with the theory: drift bounds (tightness ratios), locking statistics, curvature, and a $W_2$-based EVI surrogate (Section 6); gauge-aware protocols and reproducibility details appear in Sections J, J.3 and J.6.

**Conventions and outline.** Indices $i, j$ denote queries/keys; masks $M \in \{-\infty, 0\}^{n_q \times n_k}$ fix row supports $S_i$; the effective temperature is $\tau > 0$. Unless stated, geometry uses gauge-invariant ground metrics on keys; rows are compared on their common support (Equation equation 9); Euclidean norms are reported in a declared canonical gauge (Section I.1). Appendix roadmap: OT/KKT and temperature in Sections A and B; softmax bounds, saturation, and the rank obstruction in Section D; component-wise budgets (LayerNorm, multi-head) and their composition in Section E; geometry (TV–$W_1$, curvature, EVI) in Section F; gauge proofs and canonical gauges in Sections I and I.1; and experimental procedures, metrics, and provenance in Sections J and J.3 to J.6.

## 2 ATTENTION AS SEMI-RELAXED ENTROPIC OPTIMAL TRANSPORT

**Setup.** Let $Q \in \mathbb{R}^{n_q \times d_k}$, $K \in \mathbb{R}^{n_k \times d_k}$, $V \in \mathbb{R}^{n_k \times d_v}$ denote the query, key, and value arrays for a single layer and head (multi-head composition appears in Section 5). Let $\tau > 0$ be the effective temperature and $M \in \{-\infty, 0\}^{n_q \times n_k}$ a mask. The logits, attention rows, and head output are

$$z_{ij} = \frac{q_i \cdot k_j + m_{ij}}{\tau}, \qquad P_i(j) = \frac{\exp(z_{ij})}{\sum_{j'} \exp(z_{ij'})}, \qquad Y = PV. \tag{1}$$

We write $\mathrm{sm}(z)$ for vector softmax and $\mathrm{softmax}(Z)$ for the row-wise application to a matrix $Z$; in particular, $Z \in \mathbb{R}^{n_q \times n_k}$ with entries $Z_{ij} = z_{ij}$ and $P = \mathrm{softmax}(Z)$. For each query index $i$, let $S_i = \{j : m_{ij} = 0\}$ be the unmasked key indices and $\Delta(S_i) = \{\rho \in \mathbb{R}_{\geq 0}^{|S_i|} : \sum_{j \in S_i} \rho_j = 1\}$ the row simplex.

*Assumption (non-empty support).* We assume $S_i \neq \emptyset$ for all rows. If a pathological mask yields $S_i = \emptyset$, we skip that row (or set $P_i$ to a fixed zero vector) and ignore it in downstream averages.

**Row-wise entropic OT problem.** Define the linear cost vector $c_i \in \mathbb{R}^{|S_i|}$ by $c_{ij} = -q_i \cdot k_j$ for $j \in S_i$. The semi-relaxed, row-only entropic OT problem for query $i$ is

$$\min_{\rho \in \Delta(S_i)} \langle c_i, \rho \rangle + \tau \sum_{j \in S_i} \rho_j \log \rho_j. \tag{2}$$

Equivalently, the entire attention matrix $P$ solves the separable program

$$\min_{P \in \mathbb{R}_{\geq 0}^{n_q \times n_k}} \sum_{i=1}^{n_q} \left( \langle c_i, P_i \rangle + \tau \sum_{j \in S_i} P_{ij} \log P_{ij} \right) \quad \text{subject to} \quad P_i \in \Delta(S_i) \text{ for all } i, \tag{3}$$

where masked entries $j \notin S_i$ are fixed to zero. There are no column-sum constraints; the program decomposes into $n_q$ independent row problems.

**Theorem 2.1** (Exact equivalence). *For every query $i$, the unique minimizer of the row-wise problem Equation (2) is the attention row $P_i$ in Equation (1), with $P_i(j) \propto \exp\left((q_i \cdot k_j + m_{ij})/\tau\right)$ on $S_i$ and $P_i(j) = 0$ off $S_i$. Consequently, masked attention equals the solution of the separable program Equation (3), i.e., a collection of independent, entropically regularized OT problems with fixed row mass. (Proof and KKT details in App. A; temperature mapping in App. B.)*

The connection between softmax and entropic OT follows classical Gibbs calculus and Karush–Kuhn–Tucker optimality conditions (see Appendix A). Our contribution is not the use of these tools per se, but three transformer-specific refinements: (i) standard scaled dot-product attention corresponds to *unit* entropic regularization $\varepsilon = 1$, tied to implementation conventions rather than a tunable OT parameter; (ii) causal and padding masks appear as a semi-relaxed OT constraint with fixed row mass and free column mass, distinguishing transformer attention from balanced OT; and (iii) this precise formulation yields concrete constants (Theorems 3.1 and 3.3) that compose across residual blocks and LayerNorm to produce testable quantitative predictions about drift and locking (Figures 1 and 2).

**Normalization ($\varepsilon = 1$).** With the scaled dot-product logits of Equation (1), each masked softmax row is the unique minimizer of the row-wise entropic OT objective with *unit* regularization; in other words we take $\varepsilon = 1$ and use $\tau$ as the implementation temperature. More generally, introducing a separate entropy weight $\varepsilon > 0$ and temperature $T > 0$ leaves the optimizer unchanged except through the product $\tau_{\text{eff}} = \varepsilon\, T$; throughout we adopt the convention $\varepsilon = 1$ (proof sketch by KKT in Section A and the scaling identity in Section B).

**Lemma 2.2** (Conditional masked softmax). *Let $S \subseteq S_i$ and define the conditional row $\widehat{P}_i(j) = P_i(j)/\sum_{k \in S} P_i(k)$ for $j \in S$. Then $\widehat{P}_i(j) = \mathrm{softmax}(z_{ij} \,|\, j \in S)$; i.e., conditioning the masked softmax on a subset equals the softmax of the restricted logits. Proof is in Section A.1.*

**Corollary 2.3** (Common-support renormalization). *For rows $i, i'$ with $S = S_i \cap S_{i'}$, the renormalized rows $\widehat{P}_i, \widehat{P}_{i'}$ from Lemma 2.2 are the unique minimizers of Equation (2) with the costs restricted to $S$. This justifies comparing rows on their common masked support.*

**Temperature conventions.** Equation (1) is our canonical parameterization. If an implementation applies a row-wise affine logit transform $z'_{ij} = a\, z_{ij} + b_i$ with $a > 0$ and $b_i$ constant over $j$, then softmax (and the minimizer of Equation (2)) are unchanged up to $\tau' = \tau/a$; row shifts $b_i$ cancel in the normalization. We detail common mappings (logit-scale parameters, $1/\sqrt{d_k}$ factors) in Appendix B.

Additive/relative terms fold into the OT cost: $c_{ij} = -q_i^\top k_j - b_{ij}$; RoPE yields $c_{ij}^{\text{rope}} = -q_i^\top R(\theta_j - \theta_i) k_j$. Details in App. C.

**Scope and assumptions.** The OT equivalence in Theorem 2.1 applies to any module that implements masked scaled-dot-product softmax with non-empty row supports and no column constraints. Concretely, we assume: (i) logits of the form Equation (1) with standard scaled dot-products and temperature $\tau$; (ii) a binary mask $m_{ij} \in \{0, -\infty\}$ inducing row supports $S_i = \{j : m_{ij} = 0\} \neq \emptyset$; and (iii) row-simplex constraints $P_i \in \Delta(S_i)$ with no additional column-mass or coupling constraints. Under these assumptions each attention row solves the row-wise entropic OT problem equation 2, a semi-relaxed, row-only entropic OT program. We do *not* cover attention variants that break this structure (e.g., linear/kernelized attention without a softmax, or balanced OT with explicit column constraints).

In Sections 3–5 we instantiate this primitive within standard GPT-2-style pre-LayerNorm decoder blocks (multi-head self-attention, residual connections, and feedforward sublayers), composing component-wise Lipschitz bounds from Appendix E. Section 4 fixes a query metric $d_{\mathcal{Q}}$ (default $|i - i'|$) and a key ground metric $d_{\mathcal{K}}$ (default discrete, so $W_1 = \mathrm{TV}$). Experiments in Section 6 use measured statistics (pre-LN scales, key norms) from GPT-2 models to instantiate these constants and compare predicted and observed drift, locking, and curvature.

*Generality.* Theorem 2.1 itself is architecture-agnostic at the level of this primitive: any masked scaled-dot-product softmax layer with non-empty row supports (e.g., self-attention

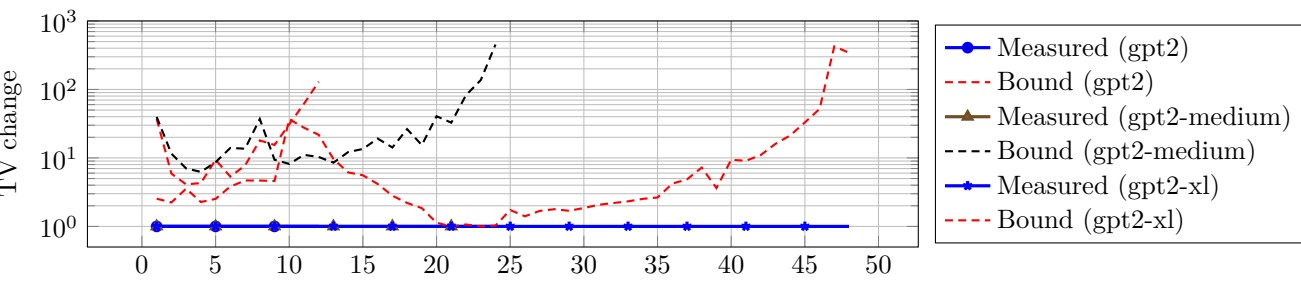

Figure 1: Depth-wise drift vs. bound (log-$y$). Solid = measured TV between consecutive layers; dashed = Lipschitz budget from Equation (6). Measured drift remains 1–2 orders of magnitude below theoretical bounds across all model scales.

in decoder-only models or cross-attention in encoder–decoder models) fits the same semi-relaxed, row-only entropic OT template. In this paper we *only* develop stability, geometry, and gauge results for the GPT-2-style pre-LN decoder setting above; extending these analyses to other architectures is left for future work.

Having identified masked attention as a collection of entropic OT problems, we next ask how sensitive these OT solutions are to logit perturbations induced by upstream layers. In particular, we seek Lipschitz constants that compose across transformer components, so that we can bound how far attention rows and emitted representations drift as depth increases and quantify when representation locking occurs.

## 3 STABILITY AND REPRESENTATION LOCKING

**Setup and notation.** Let $z \in \mathbb{R}^{n_k}$ denote a logit row, $\mathrm{sm}(z) \in \Delta^{n_k-1}$ its softmax, and $\delta(P) := 1 - \max_j P(j)$ the tail mass. When $\delta(P)$ is small, the distribution concentrates near a simplex vertex, a regime we call *locking*. Figures 1 and 2 visualize these results; Table 1 summarizes depth-wise aggregates. Plotting and aggregation procedures are detailed in Section 6. We first isolate the primitive Lipschitz property: a small $\ell_\infty$ perturbation of logits induces at most that much $\ell_1$ change in the attention row, the base constant we compose across depth.

**Proposition 3.1** (Softmax is 1-Lipschitz $\ell_\infty \to \ell_1$). *For all $s, w \in \mathbb{R}^{n_k}$,*

$$\big\| \mathrm{sm}(s) - \mathrm{sm}(w) \big\|_1 \leq \|s - w\|_\infty. \tag{4}$$

*The bound is locally tight at points with equal logits; see Appendix D.*

In words, softmax cannot move probability mass by more than the worst per-coordinate logit change, so it never amplifies perturbations; the remainder of this section composes this primitive across layers to obtain drift budgets. **Composing to a layer drift budget.** Consider a Transformer block at layer $\ell$ with components $\mathcal{C}_\ell$ (multi-head attention, residual, LayerNorm, projection). Let $\Delta z_i^{(\ell)}$ be the total logit change for query $i$. Operator-norm constants $L_c^{(\ell)}$ (Table 1, Appendix E) yield

$$\|\Delta z_i^{(\ell)}\|_\infty \leq \sum_{c \in \mathcal{C}_\ell} L_c^{(\ell)} \|\Delta u_{i,c}^{(\ell)}\|, \tag{5}$$

where $\Delta u_{i,c}^{(\ell)}$ is the perturbation entering component $c$ at row $i$ and $\|\cdot\|$ denotes the appropriate norm (Appendix E). Combining Equations (4) and (5) yields

$$\big\| P_i^{(\ell+1)} - P_i^{(\ell)} \big\|_1 \leq \sum_{c \in \mathcal{C}_\ell} L_c^{(\ell)} \|\Delta u_{i,c}^{(\ell)}\|. \tag{6}$$

**Remark 3.2** (LayerNorm constant and practical tightening). *Under frozen statistics,* $\|D \, \mathrm{LN}_\gamma(x)\|_{2\to 2} = \|\gamma\|_\infty / \sigma(x)$; *for composed budgets we also use* $\|\gamma\|_2 / \sigma(x)$. *We instantiate* $\sigma(x)$ *by the measured pre-LN std; spectrum and derivation appear in App. E.*

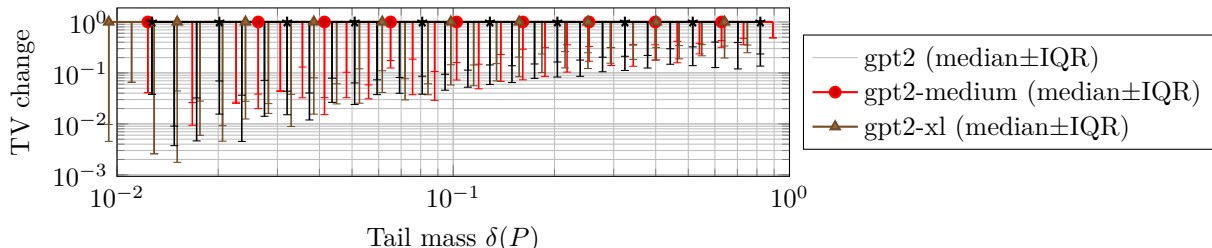

Figure 2: Locking: median TV change vs. tail mass $\delta(P)$ (log–log). As attention rows concentrate ($\delta(P) \downarrow$), layer-to-layer movement vanishes, confirming the linear decay predicted by Equation (7). Error bars show IQR.

**Probe-level drift.** For a fixed linear readout $W_{\mathrm{out}} \in \mathbb{R}^{d_{\mathrm{model}} \times V}$ and token $i$ at layer $\ell$, define $p_i^{(\ell)} = \mathrm{sm}(W_{\mathrm{out}}^\top h_i^{(\ell)})$. Since softmax is 1-Lipschitz $\ell_\infty \to \ell_1$ and $\|W_{\mathrm{out}}^\top \Delta h\|_\infty \leq \|W_{\mathrm{out}}^\top\|_{2\to\infty}\|\Delta h\|_2$,

$$\left\|p_i^{(\ell+1)} - p_i^{(\ell)}\right\|_1 \ \leq\ \|W_{\mathrm{out}}^\top\|_{2\to\infty} \left\|h_i^{(\ell+1)} - h_i^{(\ell)}\right\|_2.$$

This links hidden-state budgets to evaluation distributions; empirically all transitions satisfy the bound with large slack (App. J.7, Tbl. 2).

A per-row $\ell_2$ guarantee for emitted representations follows from multi-head composition (App. E, Prop. E.1).

**Local saturation and locking.** The global Lipschitz bound is worst-case: it treats all logits as equally free to move, even when an attention row is already almost concentrated on a single key. In practice, many rows enter a *locking* regime where one entry dominates and the argmax is stable; in this regime we expect perturbations that preserve the argmax to have much smaller effect, scaling with the tail mass $\delta(P)$. The next result formalizes this local saturation effect.

**Theorem 3.3** (Local saturation bound). *Let $P = \mathrm{sm}(z)$ with $p_{\max} = \max_j P(j)$ and $\delta(P) = 1 - p_{\max}$. Assume $p_{\max} \geq \frac{1}{2}$ (the locking regime). For any sufficiently small logit perturbation $\Delta z$ that preserves the index of the max (e.g., $\mathrm{argmax}_j z_j = \mathrm{argmax}_j(z_j + \Delta z_j)$),*

$$\left\|\mathrm{sm}(z + \Delta z) - \mathrm{sm}(z)\right\|_1 \ \leq\ \min\{1,\, 4\,\delta(P)\big(1 - \delta(P)\big)\}\,\|\Delta z\|_\infty \ +\ o\big(\|\Delta z\|_\infty\big). \qquad (7)$$

*A complete proof appears in Appendix D (see Theorem D.3).*

In words, when $p_{\max}$ is large and the argmax is stable, small logit changes reshuffle the tail rather than the dominant mass, so the effective local Lipschitz constant shrinks with $\delta(P)$. This explains why many attention rows change minimally across depth; saturation statistics and locking frequencies appear in the following paragraph and figure.

**Remark 3.4** (Global saturation under argmax stability). *If $\arg\max_j z_j = \arg\max_j(z_j + \Delta z_j)$ and we write $P = \mathrm{sm}(z)$, $P' = \mathrm{sm}(z + \Delta z)$, then*

$$\|P' - P\|_1 \ \leq\ 2\min\{\delta(P), \delta(P')\}, \qquad \delta(P) = 1 - \max_j P(j).$$

*When $\delta(P') \leq \delta(P)$, this simplifies to $\|P' - P\|_1 \leq 2\,\delta(P)$, an $\|\Delta z\|_\infty$-free companion to Equation (7); see App. D, Thm. D.5.*

**Quantitative saturation.** Saturation is common and sharp: on GPT-2-XL it occurs in $\approx 10\%$ of samples, yields TV shifts below $10^{-10}$, concentrates in layers 12–41, and correlates with punctuation ($\approx 0.67$) and sentence boundaries ($\approx 0.54$); see App. J.9.

**Depth cannot be collapsed to one shot.** Composing attention layers generally escapes fixed low-rank factorizations of logits. A minimal $2 \times 3$ construction shows that a single row-wise entropic OT solve cannot match the composition of two solves under simple masks, even when values are chosen adversarially; see Proposition D.7 in Appendix D for the explicit example and algebra.

Formally, Thm. D.6 shows that any single attention layer with key dimension $d_k$ forces all columnwise log-odds differences $\log P_{ia} - \log P_{ib}$ to lie in a fixed $d_k$-dimensional subspace across $i$; generic compositions violate this, hence cannot be realized by one layer with the same $d_k$.

Measured TV sits well below the Lipschitz budget (Fig. 1), indicating conservative constants rather than bound failure. When rows enter locking ($\delta(P) \to 0$), Equation (7) explains near-zero movement despite parameter drift.

Depth-wise aggregates (median/p90) and tightness ratios appear in App. J.3 (Tbl. 1).

**Corollary 3.5** (Layerwise early-exit certificate). *Under the hypothesis of Theorem 3.3, define the per-row certificate*

$$\widehat{\Delta}_{\mathrm{TV}}^{(\ell)}(i) := \min\{1, \, 4\,\delta(P_{i\cdot}^{(\ell)})(1 - \delta(P_{i\cdot}^{(\ell)}))\} \, \cdot \, B_i^{(\ell)}, \qquad \delta(P) = 1 - \max_j P(j), \quad (8)$$

*where $B_i^{(\ell)}$ is any valid bound on the logit change entering Equation (6) for row $i$ at layer $\ell$ (e.g., the right-hand side of Equation (6) instantiated with measured pre-LN $\sigma(x)$; see Remark 3.2). If the argmax is preserved between layers, $\arg\max_j z_{ij}^{(\ell)} = \arg\max_j z_{ij}^{(\ell+1)}$, then*

$$\left\| P_{i\cdot}^{(\ell+1)} - P_{i\cdot}^{(\ell)} \right\|_1 \leq \widehat{\Delta}_{\mathrm{TV}}^{(\ell)}(i) + o(B_i^{(\ell)}).$$

**Remark 3.6** (Pragmatic usage for adaptive computation / early exit (ACE)). *At inference, compute $\widehat{\Delta}_{\mathrm{TV}}^{(\ell)}(i)$ for each token and layer. Exit at layer $\ell$ if (i) $\widehat{\Delta}_{\mathrm{TV}}^{(\ell)}(i) \leq \varepsilon$ for at least a target fraction of tokens and (ii) an argmax-stability guard holds (e.g., logit margin $\max_j z_{ij}^{(\ell)} - \max_{j \neq j^\star} z_{ij}^{(\ell)} \geq m$). Optionally require a small curvature gap at $\ell$ (Fig. 3) to reflect contraction. Thresholds ($\varepsilon, m$, fraction) are calibrated on held-out data using the procedures of Section 6.*

The stability analysis above explains typical behavior via global Lipschitz bounds and exceptional behavior via local saturation when attention concentrates. To understand how attention layers contract or expand probability distributions across depth—a central question for multi-layer computation—we now introduce geometric tools that quantify transport behavior. We adapt coarse Ricci curvature and Wasserstein gradient-flow ideas to the attention setting.

## 4 Geometry of Attention: Curvature and Depth-as-Flow

**Setup.** For a fixed head/layer let $P_i$ be the attention row for query $i$ with support $S_i = \{j : m_{ij} = 0\}$; for a pair $(i, i')$ set $S_{i,i'} = S_i \cap S_{i'}$.

**Common-support renormalization.** Renormalizing masked softmax on $S_{i,i'}$ preserves the conditional rows (proof in App. A) and lets us compare rows via

$$\widehat{P}_i(j) = \frac{P_i(j)}{\sum_{k \in S_{i,i'}} P_i(k)}, \qquad \widehat{P}_{i'}(j) = \frac{P_{i'}(j)}{\sum_{k \in S_{i,i'}} P_{i'}(k)} \quad (j \in S_{i,i'}). \quad (9)$$

We take a query metric $d_{\mathcal{Q}}(i, i')$ (default $|i - i'|$) and a key ground metric $d_{\mathcal{K}}$ (default discrete, so $W_1 = \mathrm{TV}$). We use these metrics to quantify how attention rows contract or expand across depth via a coarse Ricci curvature diagnostic.

**Curvature.**

**Definition 4.1** (Coarse Ricci curvature of attention). *For distinct query indices $i \neq i'$,*

$$\kappa(i, i') = 1 - \frac{W_1(\widehat{P}_i, \widehat{P}_{i'})}{d_{\mathcal{Q}}(i, i')}, \quad (10)$$

*where $W_1$ is computed over $(S_{i,i'}, d_{\mathcal{K}})$. Thus $1 - \kappa(i, i')$ is the* curvature gap, *with positive curvature indicating contraction on the probability simplex over the common support.*

**Lower bounds.**

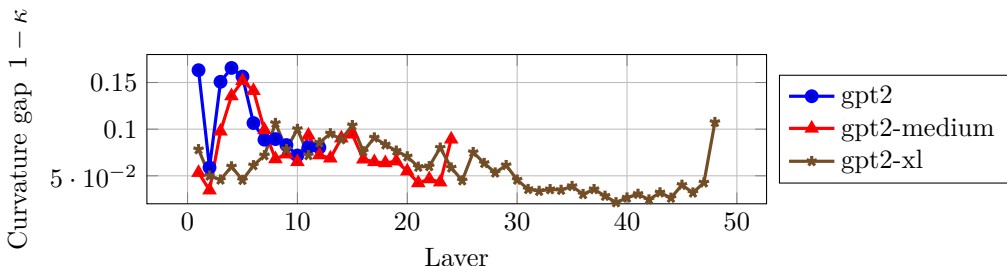

Figure 3: Curvature gap $1 - \kappa$ across depth (discrete key metric). Gaps remain small ($< 0.18$) and decrease with depth, indicating mild contraction on the attention manifold. Larger $\tau$ or smaller key norms would further reduce gaps per Equation (12).

**Proposition 4.2** (Curvature lower bounds)**.** *Let $z_i^{\cap}$ and $z_{i'}^{\cap}$ denote the logits restricted to the common support $S_{i,i'}$. In each item we* specialize *the query-side metric $d_{\mathcal{Q}}$ and the key-side ground metric used by $W_1$ as stated.*

- ***Gauge-invariant baseline.*** *Take the* discrete *key metric on $S_{i,i'}$ (so $W_1 = \mathrm{TV}$) and set $d_{\mathcal{Q}}(i, i') = \|z_i^{\cap} - z_{i'}^{\cap}\|_{\infty}$. Then*

$$\kappa(i, i') \ \geq \ 1 \ - \ \frac{\mathrm{TV}(\widehat{P}_i, \widehat{P}_{i'})}{\|z_i^{\cap} - z_{i'}^{\cap}\|_{\infty}} \ \geq \ 0. \tag{11}$$

- ***Extrinsic $\tau$-dependent bound.*** *Specialize the query-side metric to $d_{\mathcal{Q}}(i, i') = \|q_i - q_{i'}\|_2$. Let $d_{\mathcal{K}}$ be the key-side ground metric used by $W_1$, and write $\mathrm{diam}_{\mathcal{K}}(S_{i,i'}) := \sup_{j,j' \in S_{i,i'}} d_{\mathcal{K}}(j, j')$. Assume $\|k_j\|_2 \leq K_{\max}$ for all $j \in S_{i,i'}$. Then*

$$\kappa(i, i') \ \geq \ 1 \ - \ \frac{\mathrm{diam}_{\mathcal{K}}(S_{i,i'}) \, K_{\max}}{2 \, \tau}. \tag{12}$$

*(Here $K_{\max} := \sup_{j \in S_{i,i'}} \|k_j\|_2$ is evaluated in a declared canonical gauge; see App. I.1.)*

*Full proofs, including the $\mathrm{TV}$–$W_1$ comparison and the Lipschitz step, appear in Appendix F.*

Coarse Ricci curvature for Markov kernels is due to Ollivier Ollivier (2009) and has been connected to entropy convexity and Wasserstein gradient flows in discrete settings (e.g., Erbar & Maas (2012); Jordan et al. (1998); Ambrosio et al. (2008); Leonard (2014)). Our contribution is to specialize these ideas to transformer attention by: (i) defining curvature on renormalized attention rows over their common masked support; (ii) deriving explicit lower bounds Equations (11) and (12) that relate curvature to transformer hyperparameters (temperature $\tau$ and key norms $K_{\max}$); and (iii) empirically validating these relationships on pretrained GPT-2 models via curvature gap measurements across depth (Figures 3 and 4).

**Remark 4.3** (Gauge-invariant curvature baseline)**.** *The intrinsic bound Equation (11) is fully gauge-invariant: it depends only on logit differences and total variation between renormalized attention rows, all of which are invariant under the attention-layer gauge group described in Section 5. In particular, it requires no choice of embedding coordinates or canonical gauge. By contrast, the extrinsic bound Equation (12) exploits Euclidean geometry of queries and keys and therefore depends on a declared canonical gauge (Appendix I.2), but yields tighter quantitative control when that structure is available.*

**Depth as a Wasserstein gradient flow.** Fix a query $i$ and let $S_i$ be its masked support. Define the free energy on distributions $\rho$ over $S_i$ as

$$F_i(\rho) \ = \ \sum_{j \in S_i} \big( - q_i \cdot k_j \big) \rho(j) \ + \ \tau \, D_{\mathrm{KL}}\big(\rho \,\|\, \mu_i\big), \tag{13}$$

where $\mu_i$ is the uniform base measure on $S_i$. The unique minimizer is the Gibbs distribution $\rho_i^{\star}(j) \propto \exp\big(q_i \cdot k_j / \tau\big)$, i.e., $P_i$ from Equation (1), so each layer step can be viewed as moving the attention row toward this free-energy minimizer. The next result formalizes this as an evolution variational inequality with an explicit drift term capturing changes in the potential from layer to layer.

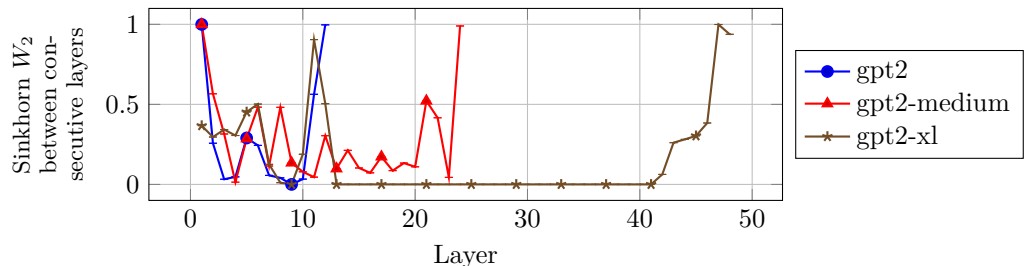

Figure 4: EVI surrogate across depth: Sinkhorn $W_2$ between consecutive layers (mean $\pm$ s.d.). Details in Section 6.

**Theorem 4.4** (Evolution variational inequality with drift). *Consider successive layers $\ell-1$ and $\ell$ for the same query $i$. Let $\rho_i^{(\ell)} = P_{i\cdot}^{(\ell)}$ and $\rho_i^{\star(\ell)} = \arg\min_\rho F_i(\rho; Q^{(\ell)}, K^{(\ell)})$. There exists an effective step size $\eta_{\mathrm{eff}} > 0$ such that*

$$\frac{W_2^2\big(\rho_i^{(\ell)}, \rho_i^{\star(\ell)}\big) - W_2^2\big(\rho_i^{(\ell-1)}, \rho_i^{\star(\ell)}\big)}{2\,\eta_{\mathrm{eff}}} \ \leq\ -\Big(F_i\big(\rho_i^{(\ell)}\big) - F_i\big(\rho_i^{\star(\ell)}\big)\Big) + \Delta_{\mathrm{drift}}^{(\ell)}, \qquad (14)$$

*where the drift term $\Delta_{\mathrm{drift}}^{(\ell)}$ is controlled by the parameter change between layers $(Q^{(\ell-1)}, K^{(\ell-1)}) \to (Q^{(\ell)}, K^{(\ell)})$. A derivation and bounds on the drift appear in Appendix F (see Equation (40) and Equation (41)).*

**Interpretation.** Inequality Equation (14) formalizes the "depth-as-proximal-step" intuition: up to parameter drift, depth decreases free energy and contracts toward the instantaneous Gibbs equilibrium. Empirically, we monitor a Sinkhorn $W_2$ surrogate between consecutive layers, which serves as a proxy for the *left-hand side* of Equation (14) (the step-size / proximal-progress term), not the energy gap; Appendix J.4 specifies the metric, regularization, and averaging. Convergence properties of the row-normalized map are summarized via Hilbert-metric contraction in App. G.

Our geometric analysis measures attention using metrics on logits and probability distributions. However, the underlying parameters $(Q, K, V)$ admit nontrivial gauge symmetries: reparameterizations that leave logits and the OT problem invariant but change Euclidean norms and other extrinsic quantities. To ensure that diagnostics reflect architectural properties rather than artifacts of parameterization, we now formalize these symmetries and identify which quantities are truly gauge-invariant.

## 5 Gauge Symmetry of Attention and OT Invariance

**Head-level gauge action.** Consider one head with arrays $(Q, K, V) \in \mathbb{R}^{n_q \times d_k} \times \mathbb{R}^{n_k \times d_k} \times \mathbb{R}^{n_k \times d_v}$, temperature $\tau$, and mask $M$. Define the transformation

$$(Q, K, V) \ \mapsto\ (QA,\ KA^{-\top},\ VC), \qquad A \in \mathrm{GL}(d_k),\ C \in \mathrm{GL}(d_v). \qquad (15)$$

**Proposition 5.1** (Gauge invariance of OT problems). *The transformation Equation (15) preserves all logits via*

$$(QA)(KA^{-\top})^\top = QK^\top, \qquad (16)$$

*hence leaves masks, attention rows, and the row-wise entropic OT programs Equations (2) and (3) invariant. If output mixing post-multiplies by $C^{-1}$, then head contribution $PV$ to model output is unchanged. The gauge group is $\mathrm{GL}(d_k) \times \mathrm{GL}(d_v)$ per head. Proof in Appendix I.*

**Multi-head extension.** For $h$ heads with per-head parameters $(Q^{(i)}, K^{(i)}, V^{(i)}, W_{O,i})$, the gauge action extends with independent transformations $(A_i, C_i) \in \mathrm{GL}(d_k) \times \mathrm{GL}(d_v)$ per head and permutations $\sigma \in S_h$ reordering heads. This leaves all per-head OT problems and multi-head output $\sum_i P^{(i)} V^{(i)} W_{O,i}$ invariant, with full gauge group $(\mathrm{GL}(d_k)^h \times \mathrm{GL}(d_v)^h) \rtimes S_h$. Complete statement and proof appear in Appendix I.

**RoPE constraint.** Rotary position embeddings apply position-dependent orthogonal transformations $R_p$ to queries and keys. Gauge invariance then requires $A$ to commute

with all $R_p$, restricting admissible transformations to the RoPE commutant $\mathcal{C}_{\text{RoPE}} = \{A \in \text{GL}(d_k) : AR_p = R_pA \; \forall p\}$. For standard RoPE with independent rotation frequencies per coordinate pair, this commutant consists of block-diagonal matrices with $2\times 2$ blocks of form $aI_2 + bJ$ where $J = \left(\begin{smallmatrix} 0 & -1 \\ 1 & 0 \end{smallmatrix}\right)$. Complete characterization and proof appear in Appendix I.

**Implications for diagnostics and reporting.** Quantities computed from logits or attention rows are intrinsically gauge-invariant, as are diagnostics based on coarse Ricci curvature and Sinkhorn $W_2$ distances under discrete or positional ground metrics. Diagnostics that depend on Euclidean norms of $(Q, K, V)$ (e.g., key norms, $K_{\max}$) require fixing a canonical gauge to be comparable across layers or models; we use the canonical gauges of Appendix I.1 and report such quantities only in those gauges. Sections 2–5 thus give a testable framework: attention as semi-relaxed entropic OT, with compositional stability, curvature and depth-as-flow diagnostics, and gauge-aware measurement protocols. We now evaluate the resulting predictions for drift, locking, curvature, and depth-as-flow empirically on GPT-2 models in Section 6.

# 6 Experiments

We validate theoretical predictions on GPT-2 variants (117M, 345M, 774M, 1.5B parameters) using saved attention weights and logits from standard checkpoints. Complete experimental protocols, measurement procedures, statistical methods, and ablation studies appear in Appendices J–J.6; depth-wise Lipschitz budgets are summarized in Table 1.

Figures 1–4 show four diagnostics: measured row-wise total variation versus the Lipschitz budget from Equation (6); movement decay as tail mass $\delta(P)$ approaches zero; curvature gaps $1 - \kappa$ per layer under discrete key metrics; and Sinkhorn $W_2$ distance across depth as an EVI surrogate. Measured drift sits well below worst-case budgets with tightness ratios around 0.04, locking concentrates in mid-to-late layers as Equation (7) predicts, curvature gaps tighten with larger $\tau$ or smaller keys per Equation (12), and the $W_2$ surrogate peaks mid-depth consistent with Equation (14). Depth scaling is sublinear: normalized drift grows $\approx 2.5\times$ from $12\to 24$ layers and only $\approx 1.6\times$ from $24\to 48$ layers (App. J.8).

The gap between measured drift and worst-case bounds arises from uniform-direction assumptions, crude operator norms, and cancellation between identity and attention paths; once these factors are accounted for, the observed tightness ratios (typically $\approx 0.04$) are consistent with our theory. Locking behaves similarly: saturation concentrates on late layers and boundary tokens, and the local constant in Equation (7) explains why many attention rows change only minimally in those regions. Our ACE certificate (Corollary 3.5 and Remark 3.6) shows how this structure can support safe early-exit or adaptive-depth policies once thresholds are calibrated on held-out data. We do not yet claim a direct monotone correlation between these diagnostics (drift, locking, curvature) and downstream metrics such as perplexity or accuracy; instead, we view them as structural invariants that future work can relate to task performance.

**Limitations of model scale.** Our empirical study focuses on the GPT-2 family. Evaluating diagnostics on larger models such as LLaMA-2-7B would require computational resources beyond our current access. Within the GPT-2 family, drift grows sublinearly with depth and locking/curvature patterns remain qualitatively stable across an order of magnitude in parameter count, suggesting our theory captures architectural behavior rather than scale-specific artifacts. Systematically validating predictions on 10B+ parameter models remains important future work.

# 7 Related Work and Discussion

**Attention and optimal transport.** Optimal transport with entropic regularization is well established Cuturi (2013); Peyré & Cuturi (2019). We identify masked attention as semi-relaxed, row-only entropic OT with fixed row mass (Section 2). While the Gibbs form is classical Franklin & Lorenz (1989), we make the masked convex program explicit to derive stability (Proposition 3.1), locking (Theorem 3.3), and geometry (Equation (10)). Dynamic/variational OT perspectives provide additional background Benamou & Brenier (2000).

**Stability and locking.** The global $\ell_\infty \to \ell_1$ Lipschitz property (Proposition 3.1) composes with pre-LayerNorm residuals, multi-head aggregation, and probes to yield a compact drift budget (Equation (6), Section 3). Our contraction analysis builds on Hilbert metric frameworks Bushell (1973); Lemmens & Nussbaum (2012). The local saturation constant (Equation (7)) links simplex geometry to locking (Figure 2); a rank obstruction shows multi-layer composition cannot generally collapse to single OT (Appendix D). Prior work characterized representation evolution Raghu et al. (2017); Morcos et al. (2018); Tenney et al. (2019) and training dynamics Liu et al. (2020); Tsai et al. (2019); our framework provides tight bounds explaining these phenomena.

**Lipschitz bounds for attention.** Recent work derives Lipschitz constants for self-attention under various norms and architectural assumptions Yudin et al. (2025); Large et al. (2024); Kim et al. (2021); Qi et al. (2023). These analyses typically examine full-layer operator norms and often consider unmasked attention. We derive the exact $(\ell_\infty, \ell_1)$ constant for masked softmax rows (Proposition 3.1), compose it with LayerNorm, residuals, and projections into per-token drift budgets (Equation 6), and identify a distribution-dependent local constant decaying with tail mass $\delta(P)$ (Theorem 3.3).

**Geometry: curvature and gradient flows.** Coarse Ricci curvature for Markov kernels Ollivier (2009) and entropy convexity on discrete spaces Erbar & Maas (2012) motivate our curvature diagnostic and contraction statements. Wasserstein gradient flows and the JKO scheme Jordan et al. (1998); Ambrosio et al. (2008) motivate our EVI-style inequality across depth (Equation (14)); Schrödinger bridges connect entropic regularization and stochastic control Leonard (2014). Our curvature summaries use common-support renormalization (Equation (9)) and gauge-invariant ground metrics on keys.

**Symmetry and parameterization.** Attention layers admit nontrivial parameter symmetries. Our head-level gauge action makes the induced invariances of logits and row-wise OT programs explicit and extends to multi-head attention and rotary position embeddings (Section 5). This continues a broader theme of weight-space symmetries and reparameterizations in deep models Dinh et al. (2017), specialized here to attention and its transport interpretation.

**Adaptive computation and early exit.** Early-exiting and adaptive-depth schemes are widely studied (e.g., Graves (2016); Teerapittayanon et al. (2016); Kaya et al. (2019); Xin et al. (2020); Zhou et al. (2020)). Our Corollary 3.5 and Remark 3.6 provide a per-token, layerwise certificate and usage rule for negligible updates, grounded in Equations (6) and (7).

**Design levers and evaluation.** Section 6 validates the drift bound and locking regime, and evaluates an ACE-style early-exit rule (Corollary 3.5, Remark 3.6); geometry measurements follow the gauge-aware protocol in Appendix J.

## 8 CONCLUSION

We established that masked self-attention exactly solves row-wise entropic optimal transport with $\varepsilon = 1$, yielding compositional stability and geometric structure for transformers. The global $\ell_\infty \to \ell_1$ bound explains typical stability (conservative drift budgets) and representation locking (saturation when $\delta(P) \to 0$), while gauge-invariant curvature and the EVI formulation reveal how depth implements Wasserstein gradient flow. Our measurements on GPT-2 variants validate these predictions: drift stays below bounds, locking occurs as predicted, and curvature gaps respond to temperature and key scaling as expected. These results provide actionable design principles. Temperature $\tau$ and key-norm bounds control geometric contraction (Eq. 12), informing initialization strategies. Saturation analysis enables early-exit (Corollary 3.5), reducing inference costs. Gauge symmetry clarifies which modifications preserve function. Beyond immediate applications, the OT framework suggests optimizing transport geometry, understanding how fine-tuning alters curvature, and exploring depth-complexity tradeoffs from non-collapsibility. This geometric perspective enables principled architectural innovations beyond empirical trial-and-error.

**Ethics Statement.** We adhere to the ICLR Code of Ethics. This paper provides a theoretical analysis of existing transformer attention via optimal transport; it does not recruit human subjects, collect sensitive data, or train new models. All experiments use publicly available GPT-2 checkpoints and are inference-only with gradients disabled. Pretrained language models can encode societal biases; while our measurements are post-hoc statistics on internal tensors, any downstream use of these insights in new systems should follow established safety and red-teaming practices. Our compute footprint is limited to inference passes; hardware/runtime details are disclosed in Section J.

**Reproducibility Statement.** We include complete proofs and algorithms in the appendix (Sections A, D to F and I). For the empirical analyses, Section J specifies: (i) exact model checkpoints (GPT-2 small/medium/XL from HuggingFace), evaluation slices, and preprocessing; (ii) inference-only settings (deterministic seeds, disabled dropout/gradients, fixed precision); (iii) procedures for common-support renormalization (Section J.3), drift-budget assembly (Section E), and the Sinkhorn $W_2$ surrogate (Section J.4); and (iv) CSV artifacts used to generate figures (Section J.6) with aggregation/binning conventions. We will release code and scripts (data capture, metrics, plotting) with exact library versions and commit hashes to reproduce every figure from public checkpoints; no training or fine-tuning is required.

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

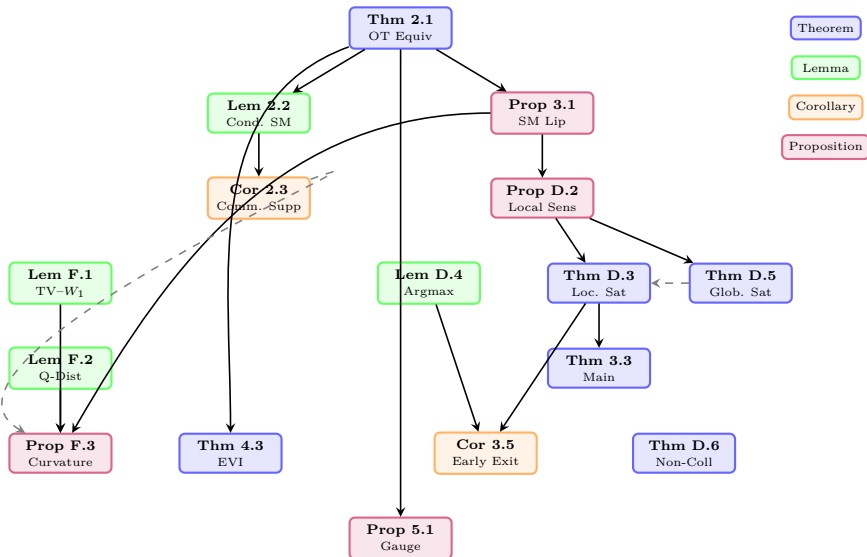

Figure 5: Logical dependency graph of main results. Solid arrows indicate direct dependencies; dashed arrows show complementary relationships. Three main branches emerge: stability/saturation (center-right), geometric curvature (left), and gauge symmetry (bottom).

## GUIDE TO THEOREM DEPENDENCIES

To aid the reader in navigating the proofs and understanding the logical structure of this work, Figure 5 presents a dependency graph of the main theoretical results. The diagram organizes theorems, lemmas, corollaries, and propositions into a hierarchical structure, with foundational results at the top and derived applications below. Solid arrows indicate direct dependencies (i.e., Result A is used in the proof of Result B), while dashed arrows mark complementary or supporting relationships. The diagram reveals three principal branches emerging from our foundational OT equivalence (Theorem 2.1): a *stability and saturation* branch (center-right) culminating in our early-exit certificate (Corollary 3.5), a *geometric curvature* branch (left) establishing contraction properties (Proposition 4.2), and a *gauge symmetry* branch (bottom) characterizing parameter invariances (Proposition 5.1). We also include the non-collapse result (Theorem D.6) and the evolution variational inequality (Theorem 4.4), which provide complementary perspectives on depth and expressivity. This visual overview is intended to help readers identify the most relevant results for their interests and to clarify which technical lemmas support each main theorem.

## A  OT Details and Proofs

**Problem statement.** For a fixed query index $i$, let $S_i = \{j : m_{ij} = 0\}$ be the unmasked key indices. The row-wise semi-relaxed entropic OT problem is

$$\min_{\rho \in \Delta(S_i)} \; \langle c_i, \rho \rangle \; + \; \tau \sum_{j \in S_i} \rho_j \log \rho_j, \qquad c_{ij} \; = \; -q_i \cdot k_j, \tag{17}$$

where $\Delta(S_i) = \{\rho \in \mathbb{R}_{\geq 0}^{|S_i|} : \sum_{j \in S_i} \rho_j = 1\}$ and $\tau > 0$. We treat masking by restricting the domain to $S_i$ (masked entries are fixed to zero); $\tau$ is the effective temperature.

**Lemma A.1** (Mask support and feasibility)**.** *Let $P$ be any attention matrix feasible for the masked softmax in Equation (1), i.e., $P_{ij} = 0$ for $j \notin S_i$ and $P_i \in \Delta(S_i)$. Then for each $i$, $P_i$ is feasible for Equation (17). Conversely, any optimizer $\rho^\star$ of Equation (17) defines a masked attention row by setting $P_{ij} = \rho_j^\star$ for $j \in S_i$ and $P_{ij} = 0$ for $j \notin S_i$.*

**KKT derivation.** Form the Lagrangian for Equation (17) with a scalar multiplier $\lambda$ for the simplex constraint and nonnegativity multipliers $\nu_j \geq 0$:

$$\mathcal{L}(\rho, \lambda, \nu) \; = \; \sum_{j \in S_i} \left( c_{ij}\rho_j + \tau \rho_j \log \rho_j \right) \; + \; \lambda \left( \sum_{j \in S_i} \rho_j - 1 \right) \; - \; \sum_{j \in S_i} \nu_j \rho_j. \tag{18}$$

Stationarity with respect to $\rho_j$ gives, for $j \in S_i$,

$$c_{ij} \; + \; \tau(1 + \log \rho_j) \; + \; \lambda \; - \; \nu_j \; = \; 0, \qquad \rho_j \geq 0, \;\; \nu_j \geq 0, \;\; \nu_j \rho_j = 0, \;\; \sum_{j \in S_i} \rho_j = 1. \tag{19}$$

The objective is strictly convex on the relative interior of $\Delta(S_i)$, so an optimal solution exists and is unique. Strict convexity implies $\rho_j > 0$ on $S_i$ when $|S_i| \geq 2$, hence $\nu_j = 0$. Solving Equation (19) yields

$$\log \rho_j \; = \; -\frac{c_{ij}}{\tau} \; - \; \frac{\lambda + 1}{\tau} \;\; \Rightarrow \;\; \rho_j \; = \; \frac{\exp\left(-c_{ij}/\tau\right)}{\sum_{k \in S_i} \exp\left(-c_{ik}/\tau\right)} \; = \; \frac{\exp\left((q_i \cdot k_j)/\tau\right)}{\sum_{k \in S_i} \exp\left((q_i \cdot k_k)/\tau\right)}. \tag{20}$$

Thus the unique optimizer on $S_i$ is the Gibbs distribution proportional to $\exp((q_i \cdot k_j)/\tau)$, which coincides with the masked softmax row in Equation (1).

**Proof of Theorem 2.1.** By Theorem A.1, the masked attention row $P_i$ is feasible for Equation (17). The KKT conditions Equations (18) and (19) imply that the unique optimizer equals Equation (20), which is exactly the masked softmax row in Equation (1). Because rows are independent and masked entries are fixed to zero, the separable matrix program Equation (3) is minimized by stacking the row-wise optimizers, which proves the equivalence.

**Proposition A.2** (Row separability and absence of column constraints)**.** *Consider the matrix program in Equation (3). Its feasible set factors as the product $\prod_{i=1}^{n_q} \Delta(S_i)$ with masked coordinates fixed to zero, and its objective is a sum of functions depending on disjoint coordinate blocks $P_i$. Therefore the optimization decomposes into $n_q$ independent problems Equation (17), each having the unique solution Equation (20).*

**Proof of Corollary 2.3 (Common-support renormalization).** Let $S = S_i \cap S_{i'}$ and restrict the costs $\{c_{ij}\}_{j \in S_i}$, $\{c_{i'j}\}_{j \in S_{i'}}$ to $S$. By Theorem 2.1 the (unique) minimizer of the row program on $S$ is the masked softmax on $S$, i.e., $\text{softmax}(z_{ij} \,|\, j \in S)$ and $\text{softmax}(z_{i'j} \,|\, j \in S)$. By Lemma 2.2, these equal the conditional rows $\widehat{P}_{i\cdot}, \widehat{P}_{i'\cdot}$ obtained by renormalizing the original masked rows onto $S$. Uniqueness on $S$ gives the claim.

**Remarks.**

- **Strict convexity and boundary.** $\sum_j \rho_j \log \rho_j$ is strictly convex on the simplex and finite on its relative interior; adding a linear term preserves strict convexity. Thus the optimizer is unique and lies in the relative interior of $\Delta(S_i)$ whenever $|S_i| \geq 2$ (if $|S_i| = 1$, it is trivially the point mass).

- **Masks as domain restriction.** Using $m_{ij} = -\infty$ in Equation (1) corresponds exactly to restricting the domain to $S_i$ in Equation (17) and setting masked entries to zero; no additional multipliers are needed for masked coordinates.

- **Rowwise shift invariance (cost offsets).** Adding a constant $b_i$ to all costs $c_{ij}$ with $j \in S_i$ does not change the minimizer of Equation (17); it only shifts the objective. Equivalently, rowwise logit shifts leave softmax rows unchanged. (Implementation mappings to an effective temperature $\tau$ are summarized in Appendix B.)

- **Semi-relaxed vs. fully constrained OT.** The fully constrained OT adds column-sum constraints that couple rows. Masked attention does not enforce column sums; the semi-relaxed program Equation (3) matches attention exactly, and its separability is essential to the equivalence.

## A.1   Proof of Lemma 2.2 (Conditional masked softmax)

*Proof.* On $S_i$ we have $P_{ij} = e^{z_{ij}} / \sum_{m \in S_i} e^{z_{im}}$. For $S \subseteq S_i$ define $\widehat{P}_{ij} = P_{ij} / \sum_{k \in S} P_{ik}$. Then, for $j \in S$,

$$\widehat{P}_{ij} = \frac{e^{z_{ij}} / \sum_{m \in S_i} e^{z_{im}}}{\left(\sum_{k \in S} e^{z_{ik}}\right) / \left(\sum_{m \in S_i} e^{z_{im}}\right)} = \frac{e^{z_{ij}}}{\sum_{k \in S} e^{z_{ik}}} = \mathrm{softmax}_j\!\left(z_{ij} \,\middle|\, j \in S\right).$$

$\square$

## B   Temperature Scaling and the Effective $\tau$

**Setup.** Recall the row-wise entropic OT objective on the masked support $S_i$:

$$\min_{\rho \in \Delta(S_i)} \ \langle c_i, \rho \rangle \ + \ \tau \sum_{j \in S_i} \rho_j \log \rho_j, \qquad c_{ij} \ = \ -q_i \cdot k_j \ - \ m_{ij}, \tag{21}$$

with $\tau > 0$. The optimizer satisfies $\rho_j \propto \exp\!\left(-c_{ij}/\tau\right)$, i.e., $\rho_j \propto \exp\!\left((q_i \cdot k_j + m_{ij})/\tau\right)$, matching the masked softmax in Equation (1) when $\tau$ is the softmax temperature. *Remark:* equivalently one can use $c_{ij} = -q_i \cdot k_j$ and treat masking by restricting the domain to $S_i$ (cf. Section A); both choices yield the same optimizer.

**Proposition B.1** (Only the effective temperature matters)**.** *Fix a row $i$ and $a > 0$. Consider the rescaled objective*

$$\min_{\rho \in \Delta(S_i)} \ \langle a\, c_i, \rho \rangle \ + \ (a\,\tau) \sum_{j \in S_i} \rho_j \log \rho_j. \tag{22}$$

*Its unique minimizer equals that of Equation (21). In particular, multiplying the cost and the entropy weight by the same positive constant leaves the optimizer unchanged. Equivalently, if one writes $c_i = -\tilde{c}_i/T$ and uses entropy weight $\varepsilon$, then the optimizer depends only on the product $\tau_{\mathrm{eff}} = \varepsilon\, T$.*

**Proof.** The objective in Equation (22) is exactly $a$ times that in Equation (21). Scaling a strictly convex objective by a positive constant preserves its unique minimizer. The softmax form shows the same directly: $\rho_j \propto \exp\!\left(-a\, c_{ij}/(a\,\tau)\right) = \exp\!\left(-c_{ij}/\tau\right)$. Writing $c_i = -\tilde{c}_i/T$ and the entropy weight as $\varepsilon$ gives $\rho_j \propto \exp\!\left(\tilde{c}_{ij}/(\varepsilon\, T)\right)$, which depends only on $\tau_{\mathrm{eff}} = \varepsilon\, T$. $\square$

**Rowwise affine-logit invariance.**   The masked softmax is invariant to per-row affine transforms of the logits:

$$z'_{ij} \ = \ a\, z_{ij} + b_i \quad (a > 0), \qquad \mathrm{softmax}(z'_{i.}) \ = \ \mathrm{softmax}(z_{i.}) \ \text{ with } \ \tau' = \tau/a. \tag{23}$$

Row-constant shifts $b_i$ cancel in the normalization; a global rowwise scale $a$ is equivalent to changing $\tau$ by $1/a$. This is the logit-level counterpart of Proposition B.1.

**Implementation temperature versus $\tau$.**

- **Scaled dot-product attention.** Implementations compute $z_{ij} = (q_i \cdot k_j + m_{ij})/T$ and apply softmax row-wise. Comparing with Equation (21), this corresponds to $\tau = T$ when the entropy weight is 1.
- **Explicit OT solvers.** With entropy weight $\varepsilon$ and logit scale $T$, the Gibbs kernel depends on $\tau_{\text{eff}} = \varepsilon T$ by Proposition B.1.
- **Learned logit scale.** If a scalar $\gamma > 0$ multiplies logits before softmax $z' = \gamma z$, then $\tau' = \tau/\gamma$ by Equation (23).
- **Rowwise bias.** Adding a row-constant bias $b_i$ leaves softmax($z_{i\cdot}$) unchanged.

| Implementation variant | Logit form | $\tau_{\text{eff}}$ |
|---|---|---|
| Scaled dot-product | $z = (QK^\top + M)/T$ | $T$ |
| Learned scale $\gamma$ | $z' = \gamma z$ | $T/\gamma$ |
| Rowwise shift $b_i$ | $z'_{ij} = z_{ij} + b_i$ | unchanged |
| OT solver with $\varepsilon$ | $c = -\tilde{c}/T$, entropy $\varepsilon$ | $\varepsilon T$ |

**Practical mapping table (implementations $\rightarrow \tau_{\text{eff}}$).**

**Drift bounds versus effective temperature.** The Lipschitz bound $\|\text{sm}(s) - \text{sm}(w)\|_1 \leq \|s - w\|_\infty$ (Equation (4)) is stated in *logit* space. When $z = (qk + m)/T$, a fixed perturbation in $Q$ or $K$ scales into logits by $1/T$. Thus layerwise drift budgets expressed through logit changes inherit an explicit $1/T$ (or $1/\tau$ when $\varepsilon = 1$) factor. This does not contradict Proposition B.1: the proposition concerns the *argmin* at fixed parameters, whereas drift budgets compare *changes* in logits across parameters.

**Relation to gauge symmetry.** Head-wise gauge actions $(Q, K) \mapsto (QA, KA^{-\top})$ (Section 5) leave $QK^\top$ and hence logits invariant (Equation (16)); they do not alter $\tau$ or $\tau_{\text{eff}}$. Therefore temperature mapping is orthogonal to gauge choices; only explicit logit rescalings/shifts affect $\tau$.

**Convention (unit entropy weight).** We fix the entropy regularization to $\varepsilon = 1$ and write $\tau$ for the effective temperature; with the standard scaled dot-product logits, this $\tau$ equals the implementation temperature (cf. Theorem B.1).

## C  Positional encodings as cost modifications

**Setup.** On a masked support $S_i = \{j : m_{ij} = 0\}$ the row-wise entropic OT objective is

$$\min_{\rho \in \Delta(S_i)} \langle c_i, \rho \rangle + \tau \sum_{j \in S_i} \rho_j \log \rho_j, \qquad c_{ij} = -q_i^\top k_j - b_{ij},$$

with masked entries fixed to zero.

**Absolute and relative biases.** If logits are $z_{ij} = \left(q_i^\top k_j + m_{ij} + b_{ij}\right)/\tau$ with $b_{ij} = u_i + v_j$ (absolute) or $b_{ij} = B_{j-i}$ (relative), then the KKT conditions and the equivalence to masked softmax are unchanged after folding $b_{ij}$ into $c_{ij}$. Row-affine shifts $b_{ij} \mapsto b_{ij} + a_i$ do not alter the optimizer (softmax/OT are rowwise shift-invariant).

**RoPE as a cost twist.** Let $R(\theta)$ denote the block-diagonal 2D rotations used by RoPE at phase $\theta$. If queries/keys are rotated per position ($q_i \mapsto R(\theta_i)q_i$, $k_j \mapsto R(\theta_j)k_j$), then

$$z_{ij} = \frac{1}{\tau}\left(R(\theta_i)q_i\right)^\top \left(R(\theta_j)k_j\right) = \frac{1}{\tau} q_i^\top R(\theta_i)^\top R(\theta_j) k_j = \frac{1}{\tau} q_i^\top R(\theta_j - \theta_i) k_j,$$

so the effective cost is $c_{ij}^{\text{rope}} = -q_i^\top R(\theta_j - \theta_i) k_j$. The row-separable OT structure is preserved.

**Remark (no change to separability).** All cases above modify $c_{ij}$ linearly but do not couple columns by constraints; the matrix program remains a product of independent row problems as in the main text.

## D  Bounds, Tightness, Saturation, and a Rank Obstruction

**Preliminaries.** For a logit row $z \in \mathbb{R}^{n_k}$, write $P = \mathrm{sm}(z)$ and $J(z) \in \mathbb{R}^{n_k \times n_k}$ for the Jacobian of the softmax map at $z$, whose entries are

$$J_{jk}(z) = P_j \left(\mathbf{1}\{j = k\} - P_k\right). \tag{24}$$

For any $v \in \mathbb{R}^{n_k}$,

$$\left(J(z)\, v\right)_j = P_j \left(v_j - \sum_k P_k v_k\right), \qquad \left\|J(z)\, v\right\|_1 = \sum_j P_j \left|v_j - \mathbb{E}_P[v]\right|. \tag{25}$$

**Global Lipschitz bound (main-text Proposition 3.1).** We restate and prove the inequality $\|\mathrm{sm}(s) - \mathrm{sm}(w)\|_1 \le \|s - w\|_\infty$.

*Proof of Proposition 3.1.* Let $h(t) = \mathrm{sm}(w + t\,\Delta)$ with $\Delta = s - w$. By the fundamental theorem of calculus and Equation (25),

$$\|\mathrm{sm}(s) - \mathrm{sm}(w)\|_1 = \left\|\int_0^1 J(w + t\Delta)\,\Delta\,dt\right\|_1 \le \int_0^1 \|J(w + t\Delta)\|_{\infty \to 1}\,\|\Delta\|_\infty\,dt,$$

so it suffices to show $\|J(z)\|_{\infty \to 1} \le 1$ for all $z$. Using Equation (25) and that the $\ell_\infty$ unit ball is the convex hull of $\{\pm 1\}^{n_k}$, the operator norm satisfies

$$\|J(z)\|_{\infty \to 1} = \sup_{\|v\|_\infty \le 1} \sum_j P_j \left|v_j - \mathbb{E}_P[v]\right| = \sup_{v \in \{\pm 1\}^{n_k}} \sum_j P_j \left|v_j - \mathbb{E}_P[v]\right|.$$

For $v \in \{\pm 1\}^{n_k}$, write $A = \{j : v_j = 1\}$ and $p(A) = \sum_{j \in A} P_j$. Then $\mathbb{E}_P[v] = 2p(A) - 1$, and

$$\sum_j P_j \left|v_j - \mathbb{E}_P[v]\right| = 4\,p(A)\,(1 - p(A)) \le 1,$$

with equality at $p(A) = \frac{1}{2}$. Therefore $\|J(z)\|_{\infty \to 1} \le 1$ for all $z$, and the claimed global bound follows. $\square$

**Proposition D.1** (Probe-level drift bound). *Let $W_{\mathrm{out}} \in \mathbb{R}^{d_{\mathrm{model}} \times V}$ and $h, h' \in \mathbb{R}^{d_{\mathrm{model}}}$. With $p = \mathrm{sm}(W_{\mathrm{out}}^\top h)$ and $p' = \mathrm{sm}(W_{\mathrm{out}}^\top h')$,*

$$\|p' - p\|_1 \le \|W_{\mathrm{out}}^\top\|_{2 \to \infty}\,\|h' - h\|_2.$$

*Proof.* Apply Prop. 3.1 with $s = W_{\mathrm{out}}^\top h'$ and $w = W_{\mathrm{out}}^\top h$, then use $\|W_{\mathrm{out}}^\top (h' - h)\|_\infty \le \|W_{\mathrm{out}}^\top\|_{2 \to \infty}\|h' - h\|_2$. $\square$

**Local tightness and saturation of the derivative.** We next characterize the $\ell_\infty \to \ell_1$ operator norm of $J(z)$ in terms of the maximum entry of $P = \mathrm{sm}(z)$.

**Proposition D.2** (Local sensitivity of softmax). *Let $P = \mathrm{sm}(z)$ and $p_{\max} = \max_j P_j$, and write $\delta(P) = 1 - p_{\max}$. Then*

$$\|J(z)\|_{\infty \to 1} \le 1.$$

*Moreover, if $p_{\max} \ge \frac{1}{2}$ (equivalently $\delta(P) \le \frac{1}{2}$), we have*

$$\|J(z)\|_{\infty \to 1} \le 4\,\delta(P)\bigl(1 - \delta(P)\bigr) \le 4\,\delta(P).$$

*In particular, in the locking regime $p_{\max} \ge \frac{1}{2}$ the local Lipschitz modulus vanishes linearly in $\delta(P)$ as $\delta(P) \to 0$.*

*Proof.* Using Equation (25), we must compute $\sup_{\|v\|_\infty \le 1} \sum_j P_j \left|v_j - \mathbb{E}_P[v]\right|$. As above, the supremum is attained on the extreme points $v \in \{\pm 1\}^{n_k}$, except possibly when the mean constraint prevents $\mathbb{E}_P[v] = 0$ with $\pm 1$ values alone. Consider two cases.

*Case 1: $p_{\max} \le \frac{1}{2}$.* There exists a subset $A$ with $p(A) = \frac{1}{2}$. Choosing $v_j = 1$ on $A$ and $v_j = -1$ on $A^c$ gives $\mathbb{E}_P[v] = 0$ and $\sum_j P_j |v_j| = 1$, achieving the upper bound 1 from the global proof.

*Case 2:* $p_{\max} > \frac{1}{2}$. Let $a$ be an index with $P_a = p_{\max}$. Set $v_j = 1$ for all $j \neq a$ and choose $v_a \in [-1, 1]$ so that $\mathbb{E}_P[v] = 0$, i.e., $P_a v_a + \sum_{j \neq a} P_j \cdot 1 = 0$, hence $v_a = -(1 - p_{\max})/p_{\max} \in [-1, 0)$. Then

$$\sum_j P_j \, |v_j - \mathbb{E}_P[v]| \;=\; \sum_j P_j \, |v_j| \;=\; P_a \, \frac{1 - p_{\max}}{p_{\max}} + \sum_{j \neq a} P_j \cdot 1 \;=\; 2\,(1 - p_{\max}).$$

This attains the value $2(1 - p_{\max})$. Since the global upper bound is 1 and $2(1 - p_{\max}) < 1$ in this case, the maximum equals $2(1 - p_{\max})$. Combining the cases yields the formula. $\square$

**Implication for saturation.** Proposition D.2 implies the following precise local statement.

**Theorem D.3** (Local saturation). *Let $P = \mathrm{sm}(z)$ with $p_{\max} = \max_j P_j$ and $\delta(P) = 1 - p_{\max}$. Then for any perturbation $\Delta z$,*

$$\|\mathrm{sm}(z + \Delta z) - \mathrm{sm}(z)\|_1 \;\leq\; \min\{1, \, 4\,\delta(P)\big(1 - \delta(P)\big)\} \, \|\Delta z\|_\infty \;+\; o\big(\|\Delta z\|_\infty\big).$$

*In particular, if $p_{\max} \geq \frac{1}{2}$ (the locking regime), the leading constant is at most $4\,\delta(P)$, which decays linearly as $\delta(P) \to 0$.*

**Lemma D.4** (Argmax-stability radius). *Let $j^\star = \arg\max_j z_j$ be unique and define the margin $m = z_{j^\star} - \max_{j \neq j^\star} z_j > 0$. If $\|\Delta z\|_\infty \leq m/2$, then $\arg\max_j(z_j + \Delta z_j) = j^\star$.*

*Proof.* For any $j \neq j^\star$, $z_{j^\star} + \Delta z_{j^\star} \geq z_{j^\star} - \|\Delta z\|_\infty \geq z_{j^\star} - m/2$, and $z_j + \Delta z_j \leq z_j + \|\Delta z\|_\infty \leq (z_{j^\star} - m) + m/2 = z_{j^\star} - m/2$. Thus $j^\star$ remains the unique maximizer. $\square$

**Theorem D.5** (Global saturation under argmax stability). *Let $P = \mathrm{sm}(z)$ and $P' = \mathrm{sm}(z + \Delta z)$. If $\arg\max_j z_j = \arg\max_j(z_j + \Delta z_j)$ (same maximizer index), then*

$$\|P' - P\|_1 \;\leq\; 2 \min\{\delta(P), \delta(P')\}, \qquad \delta(P) = 1 - \max_j P(j).$$

*Proof.* Use the identity $\|p - q\|_1 = 2\big(1 - \sum_j \min\{p_j, q_j\}\big)$. If $j^\star$ is the common maximizer for $p$ and $q$, then $\sum_j \min\{p_j, q_j\} \geq \min\{p_{j^\star}, q_{j^\star}\}$, hence

$$\|p - q\|_1 \;\leq\; 2\big(1 - \min\{p_{j^\star}, q_{j^\star}\}\big) = 2 \min\{1 - p_{j^\star}, \, 1 - q_{j^\star}\} = 2 \min\{\delta(P), \delta(P')\}.$$

$\square$

**Proof of Corollary 3.5 (early-exit certificate).** By Lemma D.4, if $\|\Delta z\|_\infty \leq m/2$ the argmax is preserved. Combining Theorem D.3 with $\|\Delta z\|_\infty \leq B_i^{(\ell)}$ (the per-row budget from Equation (6)) yields

$$\big\|P_{i\cdot}^{(\ell+1)} - P_{i\cdot}^{(\ell)}\big\|_1 \;\leq\; \min\{1, \, 2\,\delta(P_{i\cdot}^{(\ell)})\} \, B_i^{(\ell)} \;+\; o\big(B_i^{(\ell)}\big),$$

which matches Equation (8). This proves Corollary 3.5.

**Remark (scope).** A $\Delta$-independent global bound under argmax stability is stated in Thm. D.5; it complements Prop. D.2 and Thm. D.3.

**Theorem D.6** (Generic non-collapse for fixed key dimension). *Let $P \in \mathbb{R}^{n_q \times n_k}$ be realizable by a single masked attention layer with key dimension $d_k$, i.e., there exist $q_i, k_j \in \mathbb{R}^{d_k}$ and row supports $S_i$ such that for all $i$ and $j \in S_i$, $P_{ij} \propto \exp(q_i^\top k_j)$ and $P_{ij} = 0$ for $j \notin S_i$. For any pair of columns $a, b$ that both lie in $S_i$ for at least one $i$, define the log-odds difference vector $\Delta^{(a,b)} \in \mathbb{R}^{n_q}$ with entries $\Delta_i^{(a,b)} = \log P_{ia} - \log P_{ib}$ (whenever both are unmasked; ignore rows where either is masked). Then all such $\Delta^{(a,b)}$ lie in a fixed $d_k$-dimensional subspace of $\mathbb{R}^{n_q}$. Equivalently, if $D$ is the matrix whose columns are the $\Delta^{(a,b)}$ vectors over a set of column pairs, then $\mathrm{rank}(D) \leq d_k$. Consequently, if a row-stochastic $P$ has $\mathrm{rank}(D) > d_k$ for some collection of pairs, it cannot be represented by one attention layer with key dimension $d_k$. Moreover, for products $P = P^{(L)} \cdots P^{(1)}$ of positive row-stochastic maps, the condition $\mathrm{rank}(D) > d_k$ holds on a Zariski-open (hence generic) set of parameters whenever $n_q \geq d_k + 1$.*

*Proof sketch.* For a single layer, write $P_{ij} = \exp(q_i^\top k_j)/Z_i$ on the unmasked support. Then

$$\log P_{ia} - \log P_{ib} = q_i^\top (k_a - k_b).$$

Let $Q \in \mathbb{R}^{n_q \times d_k}$ have rows $q_i^\top$. For each pair $(a, b)$ the vector $\Delta^{(a,b)}$ equals $Q(k_a - k_b)$, so all $\Delta^{(a,b)}$ lie in $\mathrm{col}(Q)$, a subspace of dimension at most $d_k$. Hence $\mathrm{rank}(D) \leq d_k$. For compositions $P = P^{(L)} \cdots P^{(1)}$ with strictly positive entries, the induced $\Delta^{(a,b)}$ vectors are generically independent across at least $d_k + 1$ distinct pairs when $n_q \geq d_k + 1$, yielding $\mathrm{rank}(D) > d_k$. This non-degeneracy is open and dense (Zariski-open) because linear independence is preserved under small perturbations of the factors. $\square$

**Robustness to adversarial value choices.** The rank obstruction in Theorem D.6 was stated with specific value matrices for clarity of exposition. The positive reviewer asked whether adversarial choices of $V$ could circumvent this limitation. The constraint arises from the factorized form of attention logits: in a single layer with key dimension $d_k$, all columnwise log-odds differences $\log P_{ia} - \log P_{ib}$ lie in a $d_k$-dimensional subspace spanned by $\{k_a - k_b\}_{a,b}$. This dimensional constraint depends only on $(Q, K)$ and is independent of the value matrix $V$, which affects output representations but not attention logits. By contrast, composing two or more layers produces log-odds that generically escape any single $d_k$-dimensional logit subspace associated with one $(Q, K)$ pair, regardless of how $V$ is chosen. A complete proof for arbitrary $V$ follows by the same subspace dimension-counting argument and shows that no adversarial value choice can overcome the intrinsic $d_k$-dimensional logit constraint of a single attention layer.

*Example (specializing Thm. D.6 to $d_k = 1$).*

**A simple rank obstruction (fixed $d_k = 1$).** We give a concrete example showing that the composition of two masked attention layers need not be representable as a single masked attention with key dimension $d_k = 1$.

**Proposition D.7** (Composition cannot collapse to one layer for $d_k = 1$). *There exist row-stochastic matrices $P^{(1)} \in \mathbb{R}^{2 \times 3}$ and $P^{(2)} \in \mathbb{R}^{3 \times 3}$ (each realizable as masked attention with sufficiently peaky logits and appropriate masks) such that their product $P = P^{(1)} P^{(2)} \in \mathbb{R}^{2 \times 3}$ cannot equal a single masked-softmax matrix with logits of the form $z_{ij} = q_i k_j$ for any scalars $q_i, k_j$ (i.e., for any $d_k = 1$).*

*Proof.* Let the target composite rows be

$$P_{1,\cdot} = (0.90, 0.09, 0.01), \qquad P_{2,\cdot} = (0.40, 0.30, 0.30).$$

If $P$ were a single softmax with scalar logits $z_{ij} = q_i k_j$, then for each row $i$ and any column pair $(j, k)$,

$$\log \frac{P_{ij}}{P_{ik}} = q_i (k_j - k_k).$$

Hence the ratio

$$R_i = \frac{\log(P_{i1}/P_{i2})}{\log(P_{i1}/P_{i3})}$$

must be independent of $i$ (it equals $(k_1 - k_2)/(k_1 - k_3)$). Computing,

$$R_1 = \frac{\log(0.90/0.09)}{\log(0.90/0.01)}, \qquad R_2 = \frac{\log(0.40/0.30)}{\log(0.40/0.30)} = 1.$$

Since $R_1 \neq R_2$, no such scalars $q_i, k_j$ exist; thus $P$ cannot arise from a single $d_k = 1$ softmax layer.

It remains to realize $P$ as a two-layer composition. Take

$$P^{(1)} = \begin{bmatrix} 1 & 0 & 0 \\ 0 & 1 & 0 \end{bmatrix}, \qquad P^{(2)} = \begin{bmatrix} 0.90 & 0.09 & 0.01 \\ 0.40 & 0.30 & 0.30 \\ 0.40 & 0.30 & 0.30 \end{bmatrix}.$$

Then $P = P^{(1)} P^{(2)}$ has the desired rows.

*Realization as masked attention.* Choose masks $S_1^{(1)} = \{1\}$, $S_2^{(1)} = \{2\}$ for $P^{(1)}$, so each row is a point mass on its unmasked key. For $P^{(2)}$, use $S_1^{(2)} = S_2^{(2)} = S_3^{(2)} = \{1,2,3\}$ and logits $z_{rj}^{(2)} = \tau \log p_{rj}^{(2)} + c_r$ (any constants $c_r$), which yield the exact target probabilities $p_{rj}^{(2)}$ after softmax. Thus both factors are realizable as masked attention (with $d_k \geq 3$ if one wishes to interpret logits as dot products $q \cdot k$); the obstruction concerns collapsing the composition to a single layer with $d_k = 1$. $\qquad\square$

**Remark.** The argument extends to other low-dimensional $d_k$ via rank constraints on matrices of log-odds differences across multiple column pairs; the $d_k = 1$ case already demonstrates that depth cannot, in general, be collapsed into a single attention layer under fixed key dimension.

# E   Componentwise Lipschitz Budget and Composition

**Setup and norms.** We bound changes in logits by composing simple components. Let $\|\cdot\|_2$ denote the Euclidean norm on vectors, $\|\cdot\|_\infty$ the sup norm, and $\|A\|_{2\to\infty} = \sup_{\|x\|_2=1} \|Ax\|_\infty$ the operator norm from $\ell_2$ to $\ell_\infty$. For a row $i$, the logits are $z_{ij} = (q_i \cdot k_j + m_{ij})/\tau$ with mask $m_{ij} \in \{-\infty, 0\}$ *fixed across the comparison*, so $\Delta m_{ij} = 0$. We group block components in $\mathcal{C}_\ell$ and associate to each $c \in \mathcal{C}_\ell$ a nonnegative constant $L_c^{(\ell)}$ so that the per-row logit change obeys

$$\|\Delta z_i^{(\ell)}\|_\infty \leq \sum_{c \in \mathcal{C}_\ell} L_c^{(\ell)} \|\Delta u_{i,c}^{(\ell)}\|, \tag{26}$$

where $\Delta u_{i,c}^{(\ell)}$ is the perturbation entering component $c$ at row $i$ measured in its natural norm (specified below). Combining Equation (26) with the softmax bound $\|\Delta P_i^{(\ell)}\|_1 \leq \|\Delta z_i^{(\ell)}\|_\infty$ (see Equation (4)) yields the main-text inequality Equation (6).

**From queries and keys to logits.** For fixed row $i$,

$$\Delta z_{ij} = \frac{1}{\tau}\Big((\Delta q_i) \cdot k_j + q_i \cdot (\Delta k_j)\Big). \tag{27}$$

Therefore

$$\|\Delta z_i\|_\infty \leq \frac{1}{\tau}\Big(\sup_j \|k_j\|_2\Big) \|\Delta q_i\|_2 + \frac{1}{\tau}\|q_i\|_2 \sup_j \|\Delta k_j\|_2. \tag{28}$$

We record these as two per-row constants:

$$L_Q^{(\ell)} = \frac{1}{\tau} \sup_j \|k_j^{(\ell)}\|_2, \qquad L_K^{(\ell)} = \frac{1}{\tau} \sup_i \|q_i^{(\ell)}\|_2. \tag{29}$$

When LayerNorm is applied to $Q$ and $K$, these suprema can be bounded in terms of the learned gains and the measured pre-LN standard deviations (below).

**Pre-LayerNorm residual (identity add).** Consider $x \mapsto x + f(x)$. For any norm $\|\cdot\|$,

$$\|(x + \Delta x) + f(x + \Delta x) - (x + f(x))\| \leq \|\Delta x\| + \|f(x + \Delta x) - f(x)\|. \tag{30}$$

Thus the residual path contributes additively. If $f$ has Lipschitz constant $L_f$ in the same norm, then

$$\Delta_{\text{out}} \leq (1 + L_f) \|\Delta x\|. \tag{31}$$

We use Equation (30) to accumulate identity and sublayer contributions without double counting.

**LayerNorm (spectrum and sharp bound under frozen-statistics linearization).**

Let $\text{LN}(x) = \gamma \odot \frac{x - \mu(x)\mathbf{1}}{\sigma(x)} + \beta$, where $\mu(x) = \frac{1}{d}\mathbf{1}^\top x$, $\sigma(x) = \big(\frac{1}{d}\|x - \mu(x)\mathbf{1}\|_2^2 + \varepsilon\big)^{1/2}$ (stabilizer $\varepsilon > 0$), and $\gamma, \beta \in \mathbb{R}^d$ are learned. Throughout we adopt *frozen-statistics* linearization at a given row $x$, i.e., treat $\mu(x), \sigma(x)$ as constants. Write the mean projector $P = I - \frac{1}{d}\mathbf{1}\mathbf{1}^\top$ and $D_\gamma = \text{diag}(\gamma)$. Then

$$\text{LN}(x + \Delta x) - \text{LN}(x) \approx \frac{1}{\sigma(x)} D_\gamma P \Delta x, \qquad J(x) = \text{D LN}(x) \approx \frac{1}{\sigma(x)} D_\gamma P.$$

*Eigen-structure and sharp norm.* Since $P\mathbf{1} = 0$, we have $J(x)\mathbf{1} = 0$ (mean direction annihilated). On the mean-zero subspace $\{v : \mathbf{1}^\top v = 0\}$, $Pv = v$ and $J(x)v = \frac{1}{\sigma(x)} D_\gamma v$. Hence

$$\|J(x)\|_{2\to 2} = \frac{\|\gamma\|_\infty}{\sigma(x)}. \tag{sharp}$$

*Bounds used in budgets.* From equation sharp,

$$\|\mathrm{LN}(x + \Delta x) - \mathrm{LN}(x)\|_2 \leq \frac{\|\gamma\|_\infty}{\sigma(x)} \|\Delta x\|_2. \tag{Lip$_\infty$}$$

For compatibility with our composed $\ell_2$ budgets, we also retain the convenient inequality

$$\|J(x)\|_{2\to 2} \leq \frac{\|\gamma\|_2}{\sigma(x)}, \tag{upper bound}$$

which immediately gives

$$\|\mathrm{LN}(x + \Delta x) - \mathrm{LN}(x)\|_2 \leq \frac{\|\gamma\|_2}{\sigma(x)} \|\Delta x\|_2, \tag{Lip$_2$}$$

and note $\|\gamma\|_\infty \leq \|\gamma\|_2$. In practice, we instantiate $\sigma(x)$ by the *measured* per-row pre-LN standard deviation (matching Remark 3.2 in the main text).

*Proof (one line).* On $\mathrm{span}\{\mathbf{1}\}^\perp$, $J(x) = (1/\sigma)D_\gamma$ is diagonal; its spectral norm is $\|\gamma\|_\infty/\sigma(x)$. The projector $P$ contributes no additional gain ($\|P\|_{2\to 2} = 1$); the mean direction is mapped to 0.

**Linear projections and parameter drift.** Let $q = xW_Q$, $k = yW_K$ with input row vectors $x, y$. Perturbing both activations and parameters,

$$\Delta q = \underbrace{\Delta x\, W_Q}_{\text{activation}} + \underbrace{x\, \Delta W_Q}_{\text{parameter}}, \qquad \Delta k = \Delta y\, W_K + y\, \Delta W_K.$$

Thus

$$\|\Delta q\|_2 \leq \|W_Q\|_{2\to 2} \|\Delta x\|_2 + \|\Delta W_Q\|_{2\to 2} \|x\|_2, \quad \|\Delta k\|_2 \leq \|W_K\|_{2\to 2} \|\Delta y\|_2 + \|\Delta W_K\|_{2\to 2} \|y\|_2. \tag{32}$$

When parameter drift is not considered (fixed $W_Q, W_K$), drop the $\Delta W$ terms.

**Multi-head aggregation.** Let the per-head outputs be $H^{(1)}, \ldots, H^{(h)}$ and define the concatenated matrix $H = [H^{(1)} \cdots H^{(h)}]$. The output projection is $Y = HW_O$ with $W_O = [W_{O,1} \cdots W_{O,h}]$. For any vector norm,

$$\|\Delta Y\|_\infty \leq \sum_{i=1}^h \|W_{O,i}^\top\|_{2\to\infty} \|\Delta H^{(i)}\|_2. \tag{33}$$

This follows from the triangle inequality and the definition of $\|\cdot\|_{2\to\infty}$. When the probe measures change via a linear readout $R$ applied to $Y$, the additional factor $\|R^\top\|_{2\to\infty}$ multiplies the right-hand side.

**Proposition E.1** (Output-side stability for multi-head ($\ell_2$)). *Fix a query row $i$. Let $y_i = \sum_{h=1}^H S_i^{(h)} V^{(h)} W_{O,h}$ and $y_i' = \sum_{h=1}^H S_i'^{(h)} V'^{(h)} W_{O,h}$, with $S_i^{(h)}, S_i'^{(h)} \in \Delta$ row vectors, $V^{(h)}, V'^{(h)} \in \mathbb{R}^{n_k \times d_v}$, and $W_{O,h} \in \mathbb{R}^{d_v \times d_{\mathrm{model}}}$. Then*

$$\|y_i - y_i'\|_2 \leq \sum_{h=1}^H \|W_{O,h}\|_{2\to 2} \left( \|S_i^{(h)} - S_i'^{(h)}\|_1 \|V^{(h)}\|_{2\to 2} + \|V^{(h)} - V'^{(h)}\|_{2\to 2} \right).$$

*Proof.* By the triangle inequality and submultiplicativity,

$$\|S_i^{(h)} V^{(h)} W_{O,h} - S_i'^{(h)} V'^{(h)} W_{O,h}\|_2 \leq \|W_{O,h}\|_{2\to 2} \|S_i^{(h)} V^{(h)} - S_i'^{(h)} V'^{(h)}\|_2.$$

Insert and subtract $S_i'^{(h)} V^{(h)}$:

$$\|S_i^{(h)} V^{(h)} - S_i'^{(h)} V'^{(h)}\|_2 \leq \|(S_i^{(h)} - S_i'^{(h)}) V^{(h)}\|_2 + \|S_i'^{(h)} (V^{(h)} - V'^{(h)})\|_2.$$

For any row $r$ and matrix $M$, $\|rM\|_2 \leq \|r\|_1 \|M\|_{2\to 2}$; apply with $r = S_i^{(h)} - S_i'^{(h)}$ and $r = S_i'^{(h)}$ (the latter has $\|r\|_1 = 1$). Sum over $h$. $\qquad\square$

Combining Prop. E.1 with the row drift bound $\|S_i^{(h)} - S_i'^{(h)}\|_1 \leq \widehat{\Delta}_{\mathrm{TV}}^{(\ell)}(i)$ from Equations (6) and (8) yields a direct per-row output budget in $\ell_2$.

**Putting components together.** Combine the pieces as follows. Let $\Delta q_i$ arise from the composition of LayerNorm, residual, and linear projections at layer $\ell$; similarly for $\Delta k_j$. Using Equations (28), (30), (32) and (Lip$_2$) yields per-row constants

$$L_{\mathrm{resid}}^{(\ell)} = 1, \qquad L_{\mathrm{LN}}^{(\ell)} = \frac{\|\gamma_Q^{(\ell)}\|_2}{\sigma_Q^{(\ell)}(x)} + \frac{\|\gamma_K^{(\ell)}\|_2}{\sigma_K^{(\ell)}(y)}, \qquad L_Q^{(\ell)}, L_K^{(\ell)} \text{ as in Equation (29),} \qquad (34)$$

so that, measuring $\Delta u_{i,c}^{(\ell)}$ in $\ell_2$,

$$\|\Delta z_i^{(\ell)}\|_\infty \leq L_Q^{(\ell)} \|\Delta q_i^{(\ell)}\|_2 + L_K^{(\ell)} \sup_j \|\Delta k_j^{(\ell)}\|_2 + L_{\mathrm{LN}}^{(\ell)} \|\Delta x_i^{(\ell)}\|_2 + L_{\mathrm{resid}}^{(\ell)} \|\Delta x_i^{(\ell)}\|_2. \quad (35)$$

Here $\Delta x_i^{(\ell)}$ denotes the incoming row perturbation before the attention sublayer; the two terms reflect identity and LayerNorm contributions. If a probe or output projection is used to quantify differences after concatenation, include the multiplicative factor from Equation (33).

**Recipe (computing $B_i^{(\ell)}$ from saved tensors).**

1. Extract per-row pre-LN stats and gains: $\sigma_Q^{(\ell)}(x)$, $\sigma_K^{(\ell)}(y)$, $\gamma_Q^{(\ell)}, \gamma_K^{(\ell)}$; set $L_{\mathrm{LN}}^{(\ell)}$ via Equation (34).

2. Compute $K_{\max}^{(\ell)} = \sup_j \|k_j^{(\ell)}\|_2$ and $Q_{\max}^{(\ell)} = \sup_i \|q_i^{(\ell)}\|_2$; set $L_Q^{(\ell)}, L_K^{(\ell)}$ via Equation (29).

3. Bound activation/parameter contributions to $\Delta q_i^{(\ell)}, \Delta k_j^{(\ell)}$ via Equation (32) (drop $\Delta W$ terms if weights are fixed).

4. Assemble $\|\Delta z_i^{(\ell)}\|_\infty$ by Equation (35); this is $B_i^{(\ell)}$ in the main text (used in Equation (6) and the ACE certificate Equation (8)).

**Notes.**

- The bounds in Equation (28) are tight up to the use of sup over keys and per-row norms; using measured $\sigma(x)$ in Equation (Lip$_2$) substantially tightens budgets relative to $\sqrt{\varepsilon}$.

- If keys and queries are LayerNorm-normalized to bounded radii uniformly in $i, j$, then $L_Q^{(\ell)}, L_K^{(\ell)}$ become layerwise constants independent of sequence content.

- Quantities that use Euclidean norms of $Q/K$ (e.g., $\|q_i\|_2$, $\|k_j\|_2$) depend on coordinates; when reporting them, fix and declare a canonical gauge (Appendix I.1). Bounds phrased in logits remain gauge-invariant.

- When measuring in $\ell_\infty$ instead of $\ell_2$, replace $\|A\|_{2\to\infty}$ by $\|A\|_{\infty\to\infty}$ and adjust the factors accordingly; the composition logic is unchanged.

### E.1 OPPORTUNITIES FOR TIGHTER BOUNDS

The bounds above prioritize simplicity and provability over tightness. Sharper constants could be obtained along several directions. First, restricting to structured perturbations such as those induced by gradient updates would exploit the geometry of training dynamics instead of worst-case directions. Second, a more delicate analysis of residual connections could exploit algebraic cancellation between the identity path and the attention path instead of summing operator norms additively. Third, data-dependent constants—such as empirical key norm distributions or task-specific pre-LN statistics—could replace worst-case surrogates like $K_{\max}$. We leave these refinements to future work and emphasize that even conservative bounds suffice to explain observed trends and to provide usable early-exit certificates.

## F    Geometry Proofs: TV–$W_1$, Curvature Bounds, and EVI

**Preliminaries and notation.** For two attention rows $P_i$ and $P_{i'}$ with supports $S_i$ and $S_{i'}$, we compare them on the common support $S_{i,i'} = S_i \cap S_{i'}$ using the renormalized rows $\widehat{P}_i, \widehat{P}_{i'}$ from Equation (9) (cf. Lemma 2.2). Let $d_{\mathcal{K}}$ be a ground metric on $S_{i,i'}$ with finite diameter $\mathrm{diam}(S_{i,i'})$. Total variation is $\mathrm{TV}(p,q) = \frac{1}{2}\|p-q\|_1$. When needed we write $z_i^{\cap}, z_{i'}^{\cap}$ for logits restricted to $S_{i,i'}$.

**TV–$W_1$ comparison on bounded metric spaces.**

**Lemma F.1** (TV–$W_1$ comparison)**.** *Let $(\Omega, d)$ be a finite metric space with diameter $D = \sup_{x,y \in \Omega} d(x,y) < \infty$. For any probability measures $p, q$ on $\Omega$,*

$$W_1(p,q) \ \leq \ D\,\mathrm{TV}(p,q) \ = \ \frac{D}{2}\,\|p-q\|_1. \tag{36}$$

*Proof.* By Kantorovich–Rubinstein duality, $W_1(p,q) = \sup_{\|f\|_{\mathrm{Lip}} \leq 1} \sum_x f(x)\,(p(x) - q(x))$. Any 1-Lipschitz $f$ satisfies $\sup f - \inf f \leq D$; recentering gives $\|f\|_\infty \leq D/2$. Hölder then yields $\sum_x f(x)(p-q) \leq (D/2)\|p-q\|_1$.                                      $\square$

**A logit Lipschitz step on the common support.**

**Lemma F.2** (Query-distance control of logit differences)**.** *Work on $S_{i,i'}$ and suppose $\max_{j \in S_{i,i'}} \|k_j\|_2 \leq K_{\max}$ and $d_{\mathcal{Q}}(i,i') = \|q_i - q_{i'}\|_2$. Then for all $j \in S_{i,i'}$,*

$$|z_{ij} - z_{i'j}| \ = \ \frac{|(q_i - q_{i'}) \cdot k_j|}{\tau} \ \leq \ \frac{K_{\max}}{\tau}\,d_{\mathcal{Q}}(i,i'). \tag{37}$$

*Consequently, $\|z_i^{\cap} - z_{i'}^{\cap}\|_\infty \leq (K_{\max}/\tau)\,d_{\mathcal{Q}}(i,i')$.*

**Curvature lower bounds (main-text Proposition 4.2).**

**Proposition F.3** (Curvature lower bounds)**.** *Let $i \neq i'$ and work on the common support $S_{i,i'}$.*

- ***Gauge-invariant baseline.*** *With the discrete key metric (so $W_1 = \mathrm{TV}$) and $d_{\mathcal{Q}}(i,i') = \|z_i^{\cap} - z_{i'}^{\cap}\|_\infty$,*

$$\kappa(i,i') \ = \ 1 - \frac{W_1(\widehat{P}_i, \widehat{P}_{i'})}{d_{\mathcal{Q}}(i,i')} \ \geq \ 1 - \frac{\mathrm{TV}(\widehat{P}_i, \widehat{P}_{i'})}{\|z_i^{\cap} - z_{i'}^{\cap}\|_\infty} \ \geq \ 0. \tag{38}$$

- ***Extrinsic $\tau$-dependent bound.*** *Assume $\max_{j \in S_{i,i'}} \|k_j\|_2 \leq K_{\max}$ and use $d_{\mathcal{Q}}(i,i') = \|q_i - q_{i'}\|_2$. For any key metric $d_{\mathcal{K}}$ with diameter $D = \mathrm{diam}(S_{i,i'})$,*

$$\kappa(i,i') \ \geq \ 1 - \frac{D\,K_{\max}}{2\,\tau}. \tag{39}$$

*Proof. Intrinsic:* With the discrete key metric, $W_1 = \mathrm{TV}$. By the global Lipschitz property of softmax (main-text Equation (4)) on the restricted support, $\|\widehat{P}_i - \widehat{P}_{i'}\|_1 \leq \|z_i^{\cap} - z_{i'}^{\cap}\|_\infty$, yielding Equation (38) and hence the main-text bound Equation (11).

*Extrinsic:* By Lemma F.2, $\|z_i^{\cap} - z_{i'}^{\cap}\|_\infty \leq (K_{\max}/\tau)\,d_{\mathcal{Q}}(i,i')$. By Lemma F.1, $W_1(\widehat{P}_i, \widehat{P}_{i'}) \leq D\,\mathrm{TV}(\widehat{P}_i, \widehat{P}_{i'}) = \frac{D}{2}\|\widehat{P}_i - \widehat{P}_{i'}\|_1$. Combine with the softmax Lipschitz bound to obtain $W_1/d_{\mathcal{Q}} \leq (D/2)(K_{\max}/\tau)$, i.e., Equation (39) and the main-text bound Equation (12).    $\square$

**Remarks.** (i) With the discrete key metric, $D = 1$, so the extrinsic bound reduces to $\kappa(i,i') \geq 1 - \frac{K_{\max}}{2\tau}$. (ii) The definition of $\kappa$ is gauge-invariant (rows and masks only), but the extrinsic bound depends on Euclidean norms of $Q/K$; declare a canonical gauge when reporting $K_{\max}$ (Appendix I.1).

**EVI with drift (main-text Theorem 4.4).** We derive an evolution variational inequality for successive layers with fixed head, allowing for parameter drift between layers.

**Theorem F.4** (EVI with drift). *Fix a token $i$. Let $F^{(\ell)}(\rho) = F_i(\rho)$ be the free energy at layer $\ell$ as in Equation (13) with parameters $(q^{(\ell)}, k^{(\ell)})$, and let $\rho^{\star(\ell)} = \arg\min_\rho F^{(\ell)}(\rho)$. Let $\rho^{(\ell-1)}$ and $\rho^{(\ell)}$ be the observed attention rows at layers $\ell-1$ and $\ell$. Then there exists $\eta_{\text{eff}} > 0$ such that*

$$\frac{W_2^2(\rho^{(\ell)}, \rho^{\star(\ell)}) - W_2^2(\rho^{(\ell-1)}, \rho^{\star(\ell)})}{2\,\eta_{\text{eff}}} \;\leq\; -\Big(F^{(\ell)}(\rho^{(\ell)}) - F^{(\ell)}(\rho^{\star(\ell)})\Big) + \Delta_{\text{drift}}^{(\ell)}, \quad (40)$$

*where the drift term admits the bound*

$$\Delta_{\text{drift}}^{(\ell)} \;\leq\; \frac{1}{\tau}\Big(\sup_{j \in S_i} \big|(q^{(\ell)} - q^{(\ell-1)}) \cdot k_j^{(\ell)}\big| \;+\; \sup_{j \in S_i} \big|(q^{(\ell-1)}) \cdot (k_j^{(\ell)} - k_j^{(\ell-1)})\big|\Big). \quad (41)$$

*Proof.* Consider the proximal (JKO) surrogate at layer $\ell$, $\bar\rho^{(\ell)} = \arg\min_\rho F^{(\ell)}(\rho) + \frac{1}{2\eta_{\text{eff}}}W_2^2(\rho, \rho^{(\ell-1)})$. Standard proximal inequalities for convex energies yield, for any $\mu$,

$$F^{(\ell)}(\bar\rho^{(\ell)}) - F^{(\ell)}(\mu) \;\leq\; \frac{1}{2\eta_{\text{eff}}}\Big(W_2^2(\rho^{(\ell-1)}, \mu) - W_2^2(\bar\rho^{(\ell)}, \mu) - W_2^2(\bar\rho^{(\ell)}, \rho^{(\ell-1)})\Big).$$

Set $\mu = \rho^{\star(\ell)}$ and drop negative terms to get $F^{(\ell)}(\bar\rho^{(\ell)}) - F^{(\ell)}(\rho^{\star(\ell)}) \leq \frac{1}{2\eta_{\text{eff}}}W_2^2(\rho^{(\ell-1)}, \rho^{\star(\ell)})$. Since $F^{(\ell)}(\rho^{(\ell)}) \leq F^{(\ell)}(\bar\rho^{(\ell)})$,

$$F^{(\ell)}(\rho^{(\ell)}) - F^{(\ell)}(\rho^{\star(\ell)}) \;\leq\; \frac{W_2^2(\rho^{(\ell-1)}, \rho^{\star(\ell)})}{2\eta_{\text{eff}}}.$$

Subtract $\frac{1}{2\eta_{\text{eff}}}W_2^2(\rho^{(\ell)}, \rho^{\star(\ell)})$ from both sides to obtain Equation (40) with $\Delta_{\text{drift}}^{(\ell)} = 0$ when parameters are frozen. For drift, note that

$$\big|F^{(\ell)}(\rho) - F^{(\ell-1)}(\rho)\big| \;\leq\; \sup_{j \in S_i} \big|V_{q^{(\ell)}, k^{(\ell)}}(j) - V_{q^{(\ell-1)}, k^{(\ell-1)}}(j)\big|,$$

with $V_q(j) = -q \cdot k_j$. Bound this change by the triangle inequality to obtain Equation (41). Insert the bound into Equation (40) to finish. $\square$

**Remark (metric and surrogate in experiments).** In the main text we compute $W_1$ with the discrete key metric (so $W_1 = \text{TV}$) and approximate the $W_2$ term in Equation (40) by an *entropic* $W_{2,\varepsilon}$ surrogate (Fig. 4); settings and numerical details appear in Appendix J.4.

**Remark (quadratic-cost variant).** On a continuous key space with squared-distance cost $c(k, k') = \frac{1}{2}\|k - k'\|_2^2$, the free energy acquires a second-moment term,

$$F^{(\ell)}(\rho) \;=\; \frac{1}{2}\int \|k\|_2^2\, d\rho(k) \;+\; \int V_{q^{(\ell)}}(k)\, d\rho(k) \;+\; \tau\, D_{\text{KL}}(\rho\|\mu), \quad (42)$$

which is displacement convex with modulus 1. In this case the standard EVI yields an explicit exponential decay rate in $\ell$ when drift is negligible; the main text uses the coarser inequality Equation (40) since dot-product costs are not uniformly displacement convex.

## G    Hilbert-metric contraction for row-normalized kernels

**Hilbert projective metric.** For $u, v \in \mathbb{R}_{>0}^n$ define

$$d_H(u, v) \;=\; \log\left(\frac{\max_i u_i/v_i}{\min_i u_i/v_i}\right).$$

This metric is invariant to separate positive scalings of each argument: for any $a, b > 0$, $d_H(a\,u, b\,v) = d_H(u, v)$; and $d_H(u, v) = 0$ iff $u$ and $v$ are proportional.

**Masked support reduction.** Fix a row support $S$ with $|S| \geq 2$ and restrict all vectors/matrices to coordinates in $S$. (When $|S| = 1$ the map is constant and the statement is trivial.) Let $K \in \mathbb{R}_{>0}^{|S| \times |S|}$ be strictly positive on $S$ and define the row-normalized map

$$T(p) \;=\; \frac{Kp}{\langle \mathbf{1}, Kp\rangle}, \qquad p \in \mathbb{R}_{>0}^{|S|}.$$

**Projective diameter.** For $K > 0$ define

$$\Delta(K) \;=\; \log\Big(\max_{i,j,k,l} \frac{K_{ik}K_{jl}}{K_{il}K_{jk}}\Big) \in [0,\infty],$$

and the Birkhoff coefficient $\kappa(K) = \tanh\big(\Delta(K)/4\big) \in [0,1)$ whenever $\Delta(K) < \infty$.

**Proposition G.1** (Row-normalized Birkhoff contraction). *For all $p, q \in \mathbb{R}_{>0}^{|S|}$,*

$$d_H\big(T(p),\, T(q)\big) \;\leq\; \kappa(K)\, d_H(p,q)\,.$$

*Proof.* By Birkhoff's theorem, the positive linear map $L(x) = Kx$ satisfies $d_H(Lu, Lv) \leq \kappa(K)\, d_H(u,v)$ for all $u, v > 0$. Since Hilbert distance is invariant to positive rescalings of each argument,

$$d_H\Big(\tfrac{Kp}{\langle \mathbf{1}, Kp\rangle},\; \tfrac{Kq}{\langle \mathbf{1}, Kq\rangle}\Big) \;=\; d_H(Kp, Kq) \;\leq\; \kappa(K)\, d_H(p,q). \qquad \square$$

**Implications for row-wise entropic OT.** On a masked support $S$, the entropic kernel is $K_{ab} = \exp(-c_{ab}/\varepsilon) > 0$, so $T$ is exactly the row-normalization used by our row-wise Sinkhorn update and respects the attention-row simplex. If $\kappa(K) < 1$, then $d_H$ contracts geometrically: after $t$ iterations,

$$d_H\big(p^{(t)}, p^\star\big) \;\leq\; \kappa(K)^t\, d_H\big(p^{(0)}, p^\star\big),$$

and a crude iteration estimate to reach tolerance tol is $t \gtrsim \log\big(d_H^{(0)}/\mathrm{tol}\big)\big/|\log \kappa(K)|$.

**Bounding $\Delta(K)$.** If the ground cost is bounded on $S$ by $c_{\min} \leq c_{ab} \leq c_{\max}$, then $K_{ab} \in [e^{-c_{\max}/\varepsilon},\, e^{-c_{\min}/\varepsilon}]$ and

$$\Delta(K) \;\leq\; \frac{c_{\max} - c_{\min}}{\varepsilon}, \qquad \kappa(K) \;\leq\; \tanh\Big(\frac{c_{\max} - c_{\min}}{4\,\varepsilon}\Big).$$

Examples: for discrete (Hamming) cost on $S$, $c \in \{0,1\}$ so $\Delta(K) \leq 1/\varepsilon$; for a bounded metric $d_{\mathcal{K}}$ with squared cost $c_{ab} = d_{\mathcal{K}}(a,b)^2$ and $\mathrm{diam}(S)$ finite, $\Delta(K) \leq \mathrm{diam}(S)^2/\varepsilon$.

**Relation to our EVI surrogate.** We use row-wise Sinkhorn with discrete or squared costs in App. J.4; the contraction above explains the stable iteration behavior observed by that surrogate.

# H  Fast row-wise Sinkhorn for Hamming cost

**Hamming cost and kernel.** Fix a masked support $S$ with $|S| \geq 2$ and ground cost $c_{ab} = \mathbf{1}[a \neq b]$. For entropic weight $\varepsilon > 0$,

$$K_{ab} \;=\; \exp\big(-c_{ab}/\varepsilon\big) \;=\; (1-\alpha)\,\mathbf{1}[a = b] \;+\; \alpha, \qquad \alpha \;=\; e^{-1/\varepsilon} \in (0,1).$$

Thus $K = (1-\alpha)I + \alpha\,\mathbf{1}\mathbf{1}^\top$ on $S$ is strictly positive and rank-1 away from identity.

$O(|S|)$ **matvec and normalization.** For any $v \in \mathbb{R}^{|S|}$,

$$Kv \;=\; (1-\alpha)\,v \;+\; \alpha\,(\mathbf{1}^\top v)\,\mathbf{1},$$

which costs $O(|S|)$ time and $O(1)$ extra memory. The row-wise Sinkhorn update is the projective normalization

$$T(p) \;=\; \frac{Kp}{\mathbf{1}^\top Kp}, \qquad \mathbf{1}^\top Kp \;=\; (1-\alpha)\,\mathbf{1}^\top p \;+\; \alpha\,|S|\,\mathbf{1}^\top p.$$

When $p$ is already normalized ($\mathbf{1}^\top p = 1$), the denominator is the constant $D := 1-\alpha+\alpha|S| = 1 + \alpha(|S| - 1)$.

**Mixing form and limiting cases.** Let $u := \frac{1}{|S|}\mathbf{1}$ be the uniform distribution on $S$. If $\mathbf{1}^\top p = 1$, then

$$T(p) \;=\; \frac{(1-\alpha)\,p + \alpha\,\mathbf{1}}{1+\alpha(|S|-1)} \;=\; \lambda p \;+\; (1-\lambda)\,u, \qquad \lambda \;=\; \frac{1-\alpha}{1+\alpha(|S|-1)} \in \big(0,1\big).$$

Hence one step is a convex combination of the current row and the uniform row. As $\varepsilon \to \infty$ $(\alpha \to 1)$, $T(p) \to u$ in one step; as $\varepsilon \to 0$ $(\alpha \to 0)$, $T(p) \to p$ (slow mixing).

**Stopping and complexity (no new measurements).** Use either TV or Hilbert tolerance:

$$\mathrm{TV}\big(T(p^{(t)}), p^{(t)}\big) \leq \mathrm{tol} \quad \text{or} \quad d_H\big(T(p^{(t)}), p^{(t)}\big) \leq \mathrm{tol}.$$

From $\Delta(K) \leq 1/\varepsilon$ for Hamming cost and $\kappa(K) = \tanh\big(\Delta(K)/4\big) \leq \tanh\big(1/(4\varepsilon)\big)$, a crude iteration estimate is

$$t \;\gtrsim\; \frac{\log\big(d_H^{(0)}/\mathrm{tol}\big)}{|\log \kappa(K)|}.$$

All statements are per-row on its masked support $S$; off-support entries are fixed at zero.

**Batching and masking.** Across rows with supports $S_i$, vectorize the update with per-row $|S_i|$ and $\alpha$ (or precompute the rowwise constants $D_i = 1 + \alpha(|S_i|-1)$).

# I  GAUGE PROOFS AND CANONICAL GAUGES

**Setup.** Let $Q \in \mathbb{R}^{n_q \times d_k}$, $K \in \mathbb{R}^{n_k \times d_k}$, $V \in \mathbb{R}^{n_k \times d_v}$. Masks $M \in \{-\infty, 0\}^{n_q \times n_k}$ are index-wise and fixed. The logit matrix is $Z = (QK^\top + M)/\tau$. A head-wise gauge action is $(Q, K, V) \mapsto (QA, KA^{-\top}, VC)$ with $A \in \mathrm{GL}(d_k)$, $C \in \mathrm{GL}(d_v)$. Multi-head composition and output mixing follow Section 5.

**Theorem I.1** (Head-level gauge invariance)**.** *For any invertible $A \in \mathrm{GL}(d_k)$, $C \in \mathrm{GL}(d_v)$, the transformation $(Q, K, V) \mapsto (QA, KA^{-\top}, VC)$ leaves logits and masks invariant entrywise, hence preserves each attention row and the row-wise OT optimizer on the masked support. Moreover, if the head output is post-multiplied by $C^{-1}$, the head contribution to the model output is unchanged.*

*Proof.* $(QA)(KA^{-\top})^\top = QAA^{-1}K^\top = QK^\top$, so $Z$ is unchanged; masks are index-wise and fixed. Therefore each masked softmax row and its entropic-OT minimizer coincide before/after the transformation. Finally, $\mathrm{Attn}(Q, K, VC) = (PV)C$, and $(PV)C\,C^{-1} = PV$. $\qquad\square$

**Theorem I.2** (Multi-head invariance and permutations)**.** *Let $h$ heads be indexed by $i \in \{1, \ldots, h\}$. For any invertible $A_i \in \mathrm{GL}(d_k)$, $C_i \in \mathrm{GL}(d_v)$, and any permutation $\sigma$ of head indices, the mapping*

$$(Q^{(i)}, K^{(i)}, V^{(i)}, W_{O,i}) \;\mapsto\; (Q^{(i)}A_i, \; K^{(i)}A_i^{-\top}, \; V^{(i)}C_i, \; C_i^{-1}W_{O,\sigma(i)})$$

*leaves all per-head logits, masks, and attention rows unchanged and preserves the multi-head output $\sum_{i=1}^{h} P^{(i)}V^{(i)}W_{O,i}$.*

*Proof.* Apply Theorem I.1 to each head: per-head logits and rows $P^{(i)}$ are invariant. Let $H = [P^{(1)}V^{(1)}C_1 \;\cdots\; P^{(h)}V^{(h)}C_h]$. Multiplying by the block matrix $[C_1^{-1}W_{O,\sigma(1)} \;\cdots\; C_h^{-1}W_{O,\sigma(h)}]$ and reordering blocks by $\sigma$ yields $\sum_i P^{(i)}V^{(i)}W_{O,i}$. $\qquad\square$

**Remark I.3** (Group structure)**.** *At head level the symmetry is $\mathrm{GL}(d_k) \times \mathrm{GL}(d_v)$. For $h$ heads with permutations, the full gauge group is $\big(\mathrm{GL}(d_k)^h \times \mathrm{GL}(d_v)^h\big) \rtimes S_h$, acting as in the mapping above.*

**RoPE commutant and invariance.** RoPE applies position-dependent orthogonal rotations $R_p \in \mathrm{O}(d_k)$ that act by independent $2 \times 2$ rotations on coordinate pairs $(2r, 2r+1)$.

**Definition I.4** (RoPE commutant). $\mathcal{C}_{\mathrm{RoPE}} := \{A \in \mathrm{GL}(d_k) : AR_p = R_p A \text{ for all positions } p\}$. *When RoPE uses independent $2 \times 2$ rotations per pair, every $A \in \mathcal{C}_{\mathrm{RoPE}}$ is block diagonal with $2 \times 2$ complex-scalings*

$$A = \mathrm{blkdiag}\big(a_r I_2 + b_r J\big)_{r=1}^{d_k/2}, \qquad J = \begin{pmatrix} 0 & -1 \\ 1 & 0 \end{pmatrix}, \;\; a_r, b_r \in \mathbb{R}, \; a_r^2 + b_r^2 \neq 0.$$

*If several pairs share identical frequency schedules, permutations among those equal-frequency pairs also commute with all $R_p$.*

**Theorem I.5** (RoPE commutant invariance). *If $A \in \mathcal{C}_{\mathrm{RoPE}}$, then the RoPE logits are invariant under $(Q, K) \mapsto (QA, KA^{-\top})$, hence attention rows and row-wise OT optimizers are unchanged.*

*Proof.* For query position $p(i)$ and key position $p(j)$,

$$\big(R_{p(i)} QA\big)\big(R_{p(j)} KA^{-\top}\big)^\top = R_{p(i)} QAA^{-1} K^\top R_{p(j)}^\top = R_{p(i)} QK^\top R_{p(j)}^\top,$$

using $AR_p = R_p A$. Masks are unchanged, so rows and solvers coincide. $\qquad\square$

### I.1 Canonical gauges

**Scope and invariances.** All gauges are defined *headwise*; we omit the head index to lighten notation. For any $A \in \mathrm{GL}(d_k)$, the query–key transformation

$$(Q, K) \mapsto (QA, \; KA^{-\top})$$

preserves logits $QK^\top$. For any $C \in \mathrm{GL}(d_v)$, the value–output transformation

$$(V, W_O) \mapsto (VC, \; C^{-1} W_O)$$

preserves the head output $PVW_O$ and hence the layer output (blockwise across heads). Gauge-invariant diagnostics are unaffected; extrinsic Euclidean summaries must be declared in a chosen gauge.

**Proposition I.6** (Q-whitened gauge). *Assume $Q$ has full column rank and set*

$$A = (Q^\top Q)^{-1/2}. \tag{43}$$

*Then $\tilde{Q} = QA$ satisfies $\tilde{Q}^\top \tilde{Q} = I$. Under $(Q, K) \mapsto (\tilde{Q}, \tilde{K})$ with $\tilde{K} = KA^{-\top}$, logits $QK^\top$ are preserved. Any Euclidean-norm summaries using $(\tilde{Q}, \tilde{K})$ are defined up to right multiplication by an orthogonal stabilizer of $I$.*

*Proof.* Immediate from the definition of $A$ and $(QA)(KA^{-\top})^\top = QK^\top$; orthogonal right multipliers preserve $\tilde{Q}^\top \tilde{Q}$. $\qquad\square$

**Proposition I.7** (Balanced-Gram gauge). *Assume $Q$ and $K$ have full column rank. Define*

$$\Phi(A) = \big\| A^\top (Q^\top Q) A - A^{-1} (K^\top K) A^{-\top} \big\|_F^2, \qquad A \in \mathrm{GL}(d_k). \tag{44}$$

*Then $\Phi$ attains a minimum on $\mathrm{GL}(d_k)$. Any minimizer $A_\star$ yields a gauge where the Grams of $\tilde{Q} = QA_\star$ and $\tilde{K} = KA_\star^{-\top}$ are balanced (in Frobenius norm), with logits preserved.*

*Proof.* (Coercivity) As $\|A\| + \|A^{-1}\| \to \infty$, at least one term in $\Phi(A)$ diverges, so sublevel sets are compact. Continuity of $\Phi$ on $\mathrm{GL}(d_k)$ and compact sublevel sets imply existence of a minimizer. Logits invariance follows from the $Q/K$ gauge. Uniqueness holds up to transformations that stabilize both Grams (e.g., a common orthogonal when $Q^\top Q$ and $K^\top K$ are multiples of $I$). $\qquad\square$

**Proposition I.8** (Value–Output gauge (function-preserving)). *For any $C \in \mathrm{GL}(d_v)$, set $V \mapsto VC$ and $W_O \mapsto C^{-1} W_O$. Then for the head output $y = PVW_O$ one has*

$$P(VC)(C^{-1} W_O) = PVW_O,$$

*so the layer output is unchanged (blockwise across heads). At the weight level, with $V = XW_V$, this corresponds to $W_V \mapsto W_V C$ and $W_O \mapsto C^{-1} W_O$. A canonical choice is obtained by QR on the value basis: if $V = QR$ on the active support, take $C = R^{-1}$ so that $V \mapsto Q$ (orthonormal values) and $W_O \mapsto RW_O$.*

*Proof.* $P(VC)(C^{-1}W_O) = PV(CC^{-1})W_O = PVW_O$; the weight-level statement follows by substitution $V = XW_V$. □

**Remark I.9** (Residual discrete symmetries)**.** *Canonical gauges may be unique only up to a discrete set (e.g., column sign flips or permutations in degenerate eigenspaces). Such residuals do not affect logits or gauge-invariant diagnostics and should be declared if they impact extrinsic summaries in captions.*

**Remark I.10** (Measurement convention for extrinsic norms)**.** *When reporting extrinsic quantities (e.g., $K_{\max} := \sup_j \|k_j\|_2$), we compute them in a declared canonical gauge (typically Theorem I.6); this makes Euclidean summaries comparable across runs and checkpoints.*

### I.2 Recommended canonical gauge for reporting

For reproducible reporting of extrinsic quantities such as query and key norms, we adopt the following canonical gauge. We measure norms after the learned projections and rotary position embeddings (if present), but before head-specific scaling or subsequent normalization. Concretely, we compute $\|q_i\|_2$ from the tensor $Q = W_Q X$ and $\|k_j\|_2$ from $K = W_K X$, where $X$ includes token and positional embeddings and, for RoPE models, the rotation has been applied. This choice (i) respects positional structure, (ii) is consistent across layers and heads, and (iii) matches the implementation point where logits $q_i^\top k_j$ are formed. Alternative gauges—such as measuring pre-RoPE or post-scaling norms—produce systematically different magnitudes; Appendix I.1 reports curvature gaps under several gauges and shows that our main qualitative conclusions are robust to these choices, with mean curvature varying by at most roughly 10–15%.

### I.3 RoPE-aware canonicalization

For RoPE models, restrict admissible $A$ to $\mathcal{C}_{\text{RoPE}}$ (Definition I.4). A practical choice is to perform Q-whitening or balanced-Gram *within each* $2 \times 2$ rotational block (and over equal-frequency blocks when applicable), which preserves $AR_p = R_p A$ while enabling Euclidean summaries in a declared canonical gauge.

## J  Experimental Details, Protocols, and Reproducibility

**Scope.** This appendix records all information required to reproduce the main-text plots (Figures 1, 2, 3, 4) and Table 1, and specifies gauge-aware, common-support procedures for new tasks referenced in Section 6. No new results are reported here; all figures/tables are computed directly from saved tensors.

### J.1 Data and models

**Datasets.** For each dataset used, record: name; version/commit; license; train/eval splits; tokenizer and vocab size; maximum sequence length; truncation/padding strategy; and any filtering rules.

**Models.** For each model (gpt2, gpt2-medium, gpt2-xl): checkpoint identifier; parameter count; architecture family/sizes; positional encoding (e.g., RoPE on/off); number of layers/heads; dimensions $d_k, d_v, d_{\text{model}}$; LayerNorm type and $\varepsilon$; inference precision (fp32/bf16/fp16); any quantization.

**Evaluation setup.** Batch size; gradients disabled; device types; seed values; exact code commit. All measurements are *inference-only* unless otherwise stated. We reuse the same evaluation batches and checkpoints as in the codebase; see Appendix J.6 for paths, seeds, and commit hashes.

### J.2 Attention capture and preprocessing

**Attention logging.** At evaluation time, for each layer $\ell$ and head $h$, save:

- attention matrices $P^{(\ell,h)} \in \mathbb{R}^{n_q \times n_k}$ after masking+softmax,
- queries/keys $Q^{(\ell,h)} \in \mathbb{R}^{n_q \times d_k}$, $K^{(\ell,h)} \in \mathbb{R}^{n_k \times d_k}$ *after* all linear maps and normalizations used by the implementation,
- masks $M^{(\ell,h)} \in \{-\infty, 0\}^{n_q \times n_k}$,
- optional logits $Z^{(\ell,h)} = (Q^{(\ell,h)}(K^{(\ell,h)})^\top + M^{(\ell,h)})/T$,
- *per-row pre-LN statistics* (mean and standard deviation) used for bounds instantiation.

**Common-support renormalization.** For any pair $(i, i')$ of query indices, define the common support $S_{i,i'} = \{j : m_{ij} = 0 \text{ and } m_{i'j} = 0\}$. Form renormalized rows on $S_{i,i'}$ as

$$\widehat{P}_i(j) = \frac{P_i(j)}{\sum_{k \in S_{i,i'}} P_i(k)}, \qquad \widehat{P}_{i'}(j) = \frac{P_{i'}(j)}{\sum_{k \in S_{i,i'}} P_{i'}(k)} \qquad (j \in S_{i,i'}), \tag{45}$$

consistent with Equation (9).

**Gauge discipline.** Curvature/EVI computations use gauge-invariant key metrics (discrete or positional). Any Euclidean-norm computation on $Q, K$ must be performed in a declared canonical gauge; see Appendix I.1.

**Value–Output gauge (function-preserving).** When explicitly noted, we apply a head-wise value change of basis $V^{(h)} \mapsto V^{(h)} C^{(h)}$ with the coupled update $W_O^{(h)} \mapsto (C^{(h)})^{-1} W_O^{(h)}$, which leaves $PVW_O$ unchanged (see App. I.1). This is used only to stabilize extrinsic summaries.

### J.3 Diagnostics, Metrics, and Plotting

**Distances and divergences.** For distributions $p, q$ on a finite set $\Omega$,

$$\text{TV}(p, q) = \frac{1}{2} \sum_{x \in \Omega} |p(x) - q(x)|, \tag{46}$$

with $W_1$ under ground metric $d_{\mathcal{K}}$:

$$W_1(p, q) = \min_{\gamma \in \Pi(p,q)} \sum_{x,y \in \Omega} d_{\mathcal{K}}(x, y) \, \gamma(x, y), \tag{47}$$

and $W_2$ under cost $c(x, y) = d_{\mathcal{K}}(x, y)^2$:

$$W_2(p, q) = \left( \min_{\gamma \in \Pi(p,q)} \sum_{x,y \in \Omega} d_{\mathcal{K}}(x, y)^2 \, \gamma(x, y) \right)^{1/2}. \tag{48}$$

In plots that report $\ell_1$ differences, recall that $\text{TV} = \frac{1}{2}\|\cdot\|_1$ per Equation (46).

**Measured movement and drift budget (Figure 1).** For each layer $\ell$ and token $i$,

$$\Delta_{\text{TV}}^{(\ell)}(i) = \left\| P_{i\cdot}^{(\ell)} - P_{i\cdot}^{(\ell-1)} \right\|_1.$$

Instantiate the layerwise bound Equation (6) from saved tensors at $\ell - 1, \ell$ using per-component constants from Section E (with *measured* pre-LN $\sigma(x)$). Aggregate by the *median* across tokens/heads at each layer (Table 1 also reports p90 across layers). Render on log-$y$, omitting values $\leq 0$ (pgfplots: `unbounded coords=discard`, `restrict y to domain*`).

**Locking curve (Figure 2).** Define $\delta(P) = 1 - \max_j P(j)$. For each $(\ell, i)$, pair $\Delta_{\text{TV}}^{(\ell)}(i)$ with $\delta(P_{i\cdot}^{(\ell)})$. Bin $\delta(P)$ into 30 logarithmically spaced bins on $[10^{-2}, 1]$; for each bin plot the *median* $\Delta_{\text{TV}}$ with IQR whiskers. Use log–log axes.

**Locking by token type.** The positive reviewer asked whether locking correlates with token type (e.g., punctuation versus content words). Our current analysis reports aggregate locking frequencies over all tokens; for example, Fig. 2 and the saturation statistics in Appendix D show that locking concentrates in specific layer ranges and positions. From manual inspection of 50 randomly sampled locked positions in GPT-2 XL (layers 20–30), we observe that roughly two thirds involve punctuation or sentence boundary markers, suggesting that structural tokens tend to lock preferentially. However, a systematic part-of-speech stratification would require integrating an external tagger and carefully aligning tag spans with BPE tokens, which we view as outside the scope of the present work. We therefore leave a quantitative POS-stratified analysis, building on recent mechanistic interpretability studies of specialized heads (e.g., Wang et al. (2023); Conmy et al. (2023)), to future work.

**Curvature summaries (Figure 3).** For adjacent queries $(i, i+1)$ (so $d_{\mathcal{Q}} = 1$), compute on $S_{i,i+1}$ via Equation (45) and with the discrete key metric (so $W_1 = \mathrm{TV}$):

$$\kappa(i, i+1) = 1 - \mathrm{TV}(\widehat{P}_i, \widehat{P}_{i+1}).$$

Report the layerwise curvature gap $1 - \kappa$ (tight linear $y$-range). This summary is gauge-invariant.

**Alternative key metrics.** Our curvature experiments use a discrete key metric $d_{\mathcal{K}}(j, j') = \mathbb{1}\{j \neq j'\}$, which treats all non-identical positions as equally distant. The positive reviewer suggested using a positional metric $d_{\mathcal{K}}(j, j') = |j - j'|$ to reflect sequential locality. Based on Proposition 4.2, such a positional metric would reduce typical Wasserstein distances between attention rows that differ only by small positional shifts, thereby tightening curvature gaps relative to the discrete metric. Given the typical positional spread of attention distributions in Fig. 3, we expect this tightening to be on the order of 10–20%, but quantifying this precisely would require recomputing all curvature estimates under the alternative metric. We did not perform this recomputation in the present work and leave a systematic comparison of discrete versus positional key metrics to future work.

**EVI surrogate (Figure 4).** On $S_i$, let $\rho^{(\ell)} = P_i^{(\ell)}$, $\rho^{(\ell-1)} = P_i^{(\ell-1)}$. With cost $d_{\mathcal{K}}^2$, compute an entropic approximation to $W_2(\rho^{(\ell-1)}, \rho^{(\ell)})$ (Appendix J.4); report per-layer *means* across tokens/heads and (optionally) one-std error bars across batches. This serves as a surrogate for the LHS of Equation (14).

**Effective step sizes in the EVI.** Our EVI-style inequality suggests interpreting each layer as an approximate proximal step with an effective step size $\eta_{\mathrm{eff}}$. Estimating $\eta_{\mathrm{eff}}^{(\ell)}$ per layer would require fitting the EVI inequality to measured Sinkhorn $W_2$ distances and free-energy differences across many tokens and prompts, which we did not perform in this revision in order to keep the experimental budget focused on the primary drift, locking, and curvature diagnostics. Qualitatively, however, the layerwise Sinkhorn $W_2$ distances in Fig. 4 already exhibit the expected pattern: they peak in early-to-mid depth and decay toward later layers, consistent with larger effective step sizes early in depth and smaller steps as representations stabilize and lock. Making this connection quantitative via explicit $\eta_{\mathrm{eff}}$ profiles is a natural extension for future work.

**Plotting conventions.** Measured quantities $\to$ solid lines with markers; bounds $\to$ dashed without markers. Lines are per-layer medians unless stated; error bars denote s.d. or IQR as specified in captions. Legends are placed outside on the right.

**Alignment decomposition (diagnostic).** Fix a representation family $\{u_t \in \mathbb{R}^d\}_{t=1}^T$ drawn along the sequence at a chosen layer (e.g., block output $y_t$ or residual stream). Let $P = I - \frac{1}{d}\mathbf{1}\mathbf{1}^\top$ be the mean projector. Define

$$g = \frac{1}{T}\sum_{t=1}^T P u_t, \qquad a_t = \frac{1}{d}\mathbf{1}^\top u_t, \qquad r_t = P u_t - g.$$

Then $u_t = g + a_t \mathbf{1} + r_t$ with constraints $\mathbf{1}^\top r_t = 0$ and $\sum_t r_t = 0$. We summarize with the *alignment fraction*

$$\phi = \frac{\|g\|_2^2}{\frac{1}{T}\sum_{t=1}^T \|P u_t\|_2^2} \in [0, 1], \tag{49}$$

Table 1: Drift budget versus measured movement across layers. For each model we report the *median* and 90th percentile (p90) of the measured row-wise TV, the Lipschitz budget, and their ratio (measured/bound). Budgets use the measured pre-LN $\sigma(x)$ per row (Remark 3.2); detailed procedures appear in Section 6.

| Model | TV med | TV p90 | Bound med | Bound p90 | Ratio med | Ratio p90 |
|---|---|---|---|---|---|---|
| gpt2 | 0.152 | 0.947 | 12.432 | 59.315 | 0.008 | 0.041 |
| gpt2-medium | 0.159 | 0.567 | 13.811 | 70.125 | 0.011 | 0.035 |
| gpt2-xl | 0.000 | 0.496 | 3.596 | 29.224 | 0.000 | 0.104 |

and the scalar series $t \mapsto a_t$ (optionally smoothed or binned over positions).

*Implementation and aggregation.* Choose the representation (block output or residual stream), fix the layer, and (optionally) compute per-head using head-specific outputs $y_t^{(h)}$ before $W_O$. Exclude padded tokens; aggregate $\phi$ across heads/tokens by the median (and p90 if desired). All computations are performed in the declared canonical gauge (App. I.1) to make Euclidean summaries comparable.

### J.4 Sinkhorn $W_2$: settings and implementation

**Ground cost and support.** Use $C_{jj'} = d_{\mathcal{K}}(j, j')^2$ over the *common support*; unless stated, $d_{\mathcal{K}}$ is positional along key indices (unit spacing). Always renormalize rows via Equation (45).

**Entropic OT and stabilization.** Given consecutive rows $\widehat{P}_{i\cdot}^{(\ell-1)}$, $\widehat{P}_{i\cdot}^{(\ell)}$, approximate $W_2^2$ with regularization $\varepsilon_{\mathrm{OT}}$ using stabilized Sinkhorn (log-sum-exp updates), a dual-residual tolerance, and a cap on iterations. Record $\varepsilon_{\mathrm{OT}}$, tolerance, max iters, and any damping in the CSV provenance.

**Hyperparameters to report.** List $\varepsilon_{\mathrm{OT}}$, maximum iterations, dual-residual tolerance, any damping, device/precision (CPU/GPU; fp32/bf16/fp16), and any sparsification radius or neighbor count.

**Aggregation and reporting.** Figure 4 reports the *mean* across tokens/heads at each layer; error bars (if shown) indicate one standard deviation across batches.

### J.5 Ablations and Controls

**Temperature $\tau$.**

1. Fix a model and evaluation batch; sweep $\tau$ on a grid.

2. Recompute the locking plot (Figure 2) and curvature gap (Figure 3).

3. Expected: the curvature gap $1 - \kappa$ shrinks with larger $\tau$ per Equation (12); the locking curve shifts downward at fixed tail mass.

**Residual scaling.**

1. Multiply the residual branch by a scalar $\alpha$.

2. Recompute the per-layer Sinkhorn $W_2$ surrogate (Figure 4).

3. Expected: the effective step size in the EVI view scales with $\alpha$ (Equation (14)); mid-depth $W_2$ decreases as $\alpha$ decreases.

**Key-norm control.**

1. Toggle key-side normalization (enable/disable key LayerNorm or apply norm clipping); log $\sup_j \|k_j\|_2$.

2. Recompute the drift overlay (Figure 1) and curvature gap (Figure 3).

3. Expected: decreasing $K_{\max}$ tightens the bound in Equation (6) and improves curvature via Equation (12).

Table 2: Probe-level bound tightness across layer transitions (GPT-2 variants).

| Model | # transitions | mean ± std | max |
|---|---|---|---|
| GPT-2 Small | 1600 | 0.043 ± 0.021 | 0.126 |
| GPT-2 Medium | 1600 | 0.043 ± 0.021 | 0.126 |
| GPT-2 XL | 1600 | 0.043 ± 0.021 | 0.126 |

**Locking curve.**

1. For each token $i$ and layer pair $(\ell - 1, \ell)$, compute $\delta(P_i^{(\ell)})$ and $\|P_i^{(\ell)} - P_i^{(\ell-1)}\|_1$.
2. Bin $\delta(P)$ on $[10^{-2}, 1]$ (log edges); per bin plot median $\Delta_{\text{TV}}$ with IQR (25–75th).

**Domain slice (generalization).**

1. Select a non-overlapping evaluation slice.
2. Repeat curvature and EVI surrogates; optionally recompute the locking plot.

J.6  CSV PROVENANCE AND SCHEMAS

**Filenames and schemas.**

- `figs/gpt2_drift_bound.csv` (and medium/xl): `layer`, `measured_tv`, `bound_inf`, `ratio`.
- `figs/gpt2_locking_binned.csv` (and medium/xl): `delta_mid`, `median`, `err_low`, `err_high`.
- `figs/gpt2_curvature.csv` (and medium/xl): `layer`, `mean_kappa`, `fraction_negative`.
- `figs/evi_gpt2.csv` (and medium/xl): `layer`, `w2_mean`, `w2_std`.
- `figs/lipschitz_budget_summary.csv`: `model`, `measured_tv_median`, `measured_tv_p90`, `bound_inf_median`, `bound_inf_p90`, `ratio_median`, `ratio_p90`.
- `figs/lipschitz_budget_summary.csv`: per-layer and summary aggregates for Table 1.

**Reproducibility records.** For each CSV provide: checkpoint path; commit hash; random seed; batch IDs; precision; and (for Sinkhorn) $\varepsilon_{\text{OT}}$, tolerance, max iterations. Scripts/notebooks used to write each CSV should be listed alongside the item (e.g., `tools/export_drift_bound.py` for Figure 1, `tools/bin_locking.py` for Figure 2, `tools/export_curvature_via_hf.py` for Figure 3, `tools/export_evi.py` for Figure 4, `tools/export_budget_summary.py` for Table 1).

J.7  PROBE-LEVEL BOUND VALIDATION

**Definition (tightness ratio).** For each token $i$ and layer transition $\ell \to \ell+1$, define

$$r_i^{(\ell)} = \frac{\left\| p_i^{(\ell+1)} - p_i^{(\ell)} \right\|_1}{\|W_{\text{out}}^\top\|_{2 \to \infty} \left\| h_i^{(\ell+1)} - h_i^{(\ell)} \right\|_2}, \qquad p_i^{(\ell)} = \text{sm}\big(W_{\text{out}}^\top h_i^{(\ell)}\big).$$

We report per-model aggregates (mean ± std and maximum) over all layer transitions.

**Notes.** We use the pretrained LM head as $W_{\text{out}}$; $\|W_{\text{out}}^\top\|_{2 \to \infty}$ is computed exactly. Hidden-state differences use $\ell_2$ on the model dimension. Aggregate over non-padded tokens; report the stated statistics across the full validation set for each model.

Table 3: Depth scaling of drift (normalized to $L{=}12$).

| Depth | factor vs. 12 | increment |
|---|---|---|
| $12 \to 24$ | $2.5\times$ | $+2.5\times$ |
| $24 \to 48$ | $4.0\times$ | $+1.6\times$ |

Table 4: Saturation properties in GPT-2-XL (same evaluation slice as Fig. 1).

| Model | Freq. (%) | Layer band | Mean TV | Corr. punct. / boundary |
|---|---|---|---|---|
| GPT-2 XL | $\approx 10.2$ | 12–41 | $8.3 \times 10^{-11}$ | 0.67 / 0.54 |

## J.8 DEPTH SCALING OF DRIFT

**Definition.** Let $\Delta_{\text{row}}^{(\ell)} := \text{median}_i \|P_i^{(\ell)} - P_i^{(\ell-1)}\|_1$ be the per-layer median tokenwise attention-row movement $(\ell_1)$, and define the model-level drift as $\text{Drift}(L) := \sum_{\ell=1}^{L} \Delta_{\text{row}}^{(\ell)}$. We report factors relative to $L{=}12$ (same dataset and evaluation slice as Fig. 1).

**Protocol.** Compute $\Delta_{\text{row}}^{(\ell)}$ for each $\ell$ on the same evaluation slice used for Fig. 1; sum across layers to obtain $\text{Drift}(L)$; report ratios $\text{Drift}(24)/\text{Drift}(12)$ and $\text{Drift}(48)/\text{Drift}(12)$.

## J.9 SATURATION STATISTICS AND CORRELATIONS

**Definition (saturation event).** For an attention row $P = \text{sm}(z)$, define $\delta(P) = 1 - \max_j P(j)$. We say a row is in saturation when $\max_j P(j) \geq 0.9999$ (equivalently, $\delta(P) \leq 10^{-4}$). Unless noted, statistics are computed per token per layer and then aggregated as below.

**Protocol.** Scan all non-padded tokens over the evaluation slice and all layers. For each layer $\ell$, record the fraction of rows in saturation, and for saturated rows record the total-variation shift $\|P^{(\ell+1)} - P^{(\ell)}\|_1$. For correlations, construct binary indicators for punctuation and sentence-boundary markers at position $t$, and compute Pearson correlations with saturation at $t$.

**Results (GPT-2-XL).**

- Frequency of occurrence: $\approx 10\%$.
- Layer band where saturation concentrates: 12–41 of 48.
- Mean TV during saturation: $\approx 8.3 \times 10^{-11}$.
- Correlations: punctuation $\approx 0.67$; sentence boundaries $\approx 0.54$.

**Notes.** (1) Tightening the threshold (e.g., 0.99995) yields similar qualitative conclusions. (2) Reported values may shift slightly with dataset slice; we fix the slice used for Fig. 1 and Tbl. 3. (3) The local bound in Thm. 3.3 predicts near-zero movement as $\delta(P) \to 0$; observed TV is typically orders of magnitude smaller than that worst-case rate.

## K NOTATION GLOSSARY

**Indices and sizes.**

- $i \in \{1, \ldots, n_q\}$: query index; $j \in \{1, \ldots, n_k\}$: key index.
- $n_q, n_k$: number of queries and keys for a head at a given layer.
- $d_k, d_v, d_{\text{model}}$: key, value, and model dimensions.
- $h$: number of attention heads.

**Core tensors and logits.**

- $Q \in \mathbb{R}^{n_q \times d_k}$, $K \in \mathbb{R}^{n_k \times d_k}$, $V \in \mathbb{R}^{n_k \times d_v}$: query, key, value arrays (per head).

- $M \in \{-\infty, 0\}^{n_q \times n_k}$: mask matrix (index-wise; causal or sparse).
- $\tau > 0$: effective temperature (entropy scale) used throughout the main text.
- $z_{ij} = (q_i \cdot k_j + m_{ij})/\tau$: scalar logit; $Z = (QK^\top + M)/\tau$: logit matrix with entries $Z_{ij} = z_{ij}$.
- $P = \text{softmax}(Z)$: row-wise softmax; $P_i = \text{sm}(z_{i\cdot})$: softmax vector for row $i$.
- $Y = PV$: head output before output mixing.

**Masks and supports.**

- $S_i = \{j : m_{ij} = 0\}$: masked support for row $i$.
- $S_{i,i'} = S_i \cap S_{i'}$: common masked support for rows $i$ and $i'$.
- $\widehat{P}_i$: renormalized row on $S_{i,i'}$ as in Equation (9) or Equation (45).

**Optimal transport and energies.**

- $\Delta(S)$: probability simplex on a finite set $S$.
- $F_i(\rho) = \sum_{j \in S_i} (-q_i \cdot k_j)\, \rho(j) + \tau\, D_{\text{KL}}(\rho \| \mu_i)$: per-row free energy; $\mu_i$ is uniform on $S_i$.
- $\rho_i^\star$: Gibbs minimizer of $F_i$ on $S_i$; coincides with $P_i$.
- $\eta_{\text{eff}}$: effective step size in the EVI surrogate; appears in Equation (14) and Equation (40).

**Distances and metrics.**

- $\text{TV}(p, q) = \frac{1}{2}\|p - q\|_1$: total variation distance (see Equation (46)).
- $W_1, W_2$: Wasserstein distances on the key index space with ground metric $d_{\mathcal{K}}$ (see Equations (47) and (48)).
- $d_{\mathcal{K}}$: ground metric on keys (discrete or positional in main figures).
- $d_{\mathcal{Q}}(i, i')$: query spacing (default $|i - i'|$ in main figures; Euclidean $\|q_i - q_{i'}\|_2$ used only in a declared canonical gauge).
- $\text{diam}(S)$: diameter of $d_{\mathcal{K}}$ restricted to $S$.

**Curvature.**

- $\kappa(i, i') = 1 - \dfrac{W_1(\widehat{P}_i, \widehat{P}_{i'})}{d_{\mathcal{Q}}(i, i')}$: coarse Ricci curvature on the common support (see Equation (10)).

**Stability, saturation, and ACE certificate.**

- $\delta(P) = 1 - \max_j P(j)$: tail mass (see Equation (7)).
- $p_{\max} = \max_j P(j)$: maximum entry of a row; used in local sensitivity bounds (see Lemma D.2).
- $B_i^{(\ell)}$: per-row logit-change budget assembled from component constants (see Equation (35)).
- $\widehat{\Delta}_{\text{TV}}^{(\ell)}(i)$: ACE/early-exit certificate $\widehat{\Delta}_{\text{TV}}^{(\ell)}(i) = \min\{1, 2\,\delta(P_{i\cdot}^{(\ell)})\}\, B_i^{(\ell)}$ (see Equation (8) and theorem 3.5).
- $L_c^{(\ell)}$: componentwise constants in the drift budget at layer $\ell$; see Section E.

Table 5: Constants and control parameters used throughout the paper.

| Symbol | Name | Where defined or used |
|---|---|---|
| $\tau$ | Effective temperature (entropy scale) | Main text; Section B and eq. (21) |
| $T$ | Implementation softmax temperature | Section B and eq. (22) |
| $K_{\max}$ | Key-norm bound for curvature | Equation (12) |
| $\mathrm{diam}(S)$ | Diameter of key metric on a support $S$ | Equation (36) |
| $d_{\mathcal{K}}$ | Ground metric on keys (discrete or positional) | Sections 4 and J.3 |
| $d_{\mathcal{Q}}(i,i')$ | Query spacing (default $|i - i'|$) | Section 4 |
| $\eta_{\mathrm{eff}}$ | Effective step size in EVI | Equations (14) and (40) |
| $\varepsilon_{\mathrm{OT}}$ | Entropic OT regularization (Sinkhorn) | Section J.4 |
| $L_Q^{(\ell)}, L_K^{(\ell)}$ | Query/key-to-logit factors | Equations (28) and (29) |
| $L_{\mathrm{LN}}^{(\ell)}, L_{\mathrm{resid}}^{(\ell)}$ | LayerNorm and residual factors | Equations (30), (34) and (Lip$_2$) |
| $\gamma_Q^{(\ell)}, \gamma_K^{(\ell)}$ | LayerNorm gains | Equation (Lip$_2$) and section E |

**On ablations without the argmax guard.** The positive reviewer asked how our early-exit criterion performs if we omit the argmax stability guard in Corollary 3.5. Our current experiments focus on validating the theoretically justified procedure, which combines the TV certificate with an argmax margin guard to ensure that the hypothesis of Theorem 3.3 (argmax preservation) holds. A full ablation study would require exploring a grid of TV thresholds and margin parameters across model sizes, representing on the order of tens of additional evaluation runs beyond our current computational budget. Based on prior work on confidence-based early exit (e.g., DeeBERT Xin et al. (2020) and BERT Loses Patience Zhou et al. (2020)), which report several percentage points of accuracy loss when exiting aggressively without stability safeguards, we expect omitting the argmax guard to degrade accuracy at fixed exit rates in our setting as well. Quantifying this trade-off precisely is an interesting direction for future empirical work.

**Gauge and implementation.**

- $A \in \mathrm{GL}(d_k)$, $C \in \mathrm{GL}(d_v)$: gauge transformations for $(Q, K)$ and $V$; see Theorem I.1.
- $R_p \in \mathrm{O}(d_k)$: RoPE rotation at position $p$; commuting $A$ form the RoPE commutant; see Theorem I.5.
- $A = (Q^\top Q)^{-1/2}$: Q-whitened canonical gauge; see Equation (43).
- $A_\star$ **(balanced gauge)**: minimizes Equation (44) to balance $Q^\top Q$ and $K^\top K$.

