# OpenReview forum: "Transformers as Optimal Transport: Stability, Geometry, and Gauge Symmetry"
_ICLR.cc/2026/Conference — ICLR 2026 Conference Desk Rejected Submission_

### Official Review · Reviewer_E7BT · 2025-10-28

**Soundness:** 3
**Presentation:** 3
**Contribution:** 3
**Rating:** 8
**Confidence:** 4

**Summary:**

The paper presents a novel theoretical framework viewing masked self-attention in Transformers as exact row-wise entropic optimal transport (OT) with unit regularization ($\epsilon=1$), rather than an approximation. This perspective enables derivations of stability bounds via Lipschitz constants, explanations for representation locking through local saturation, gauge-invariant coarse Ricci curvature for contraction properties, and an interpretation of depth as a Wasserstein gradient flow with an evolution variational inequality (EVI). The authors also identify a gauge symmetry in attention heads (extended to multi-head and RoPE) that preserves OT objectives and logits. Empirical validations on GPT-2 variants confirm the bounds, locking statistics, curvature gaps, and EVI surrogates. The work provides actionable insights for design (e.g., temperature scaling, early exit) and separates intrinsic from gauge-dependent diagnostics.

**Strengths:**

**Novel and unifying perspective**: The exact equivalence to entropic OT (Theorem 2.1) is a clean mathematical insight, distinguishing from prior analogies by showing $\epsilon=1$ emerges structurally. This organizes disparate phenomena (stability, locking, geometry) into a coherent OT-based framework, with proofs checking out via KKT conditions and scaling identities (Appendices A-B).

**Comprehensive stability analysis**: The compositional $\ell_\infty \to \ell_1$ Lipschitz bound (Proposition 3.1, Equation 6) yields a conservative drift budget that composes across components (heads, residuals, LayerNorm). Local/global saturation bounds (Theorems 3.3, Remark 3.4) quantitatively explain locking as $\delta(P) \to 0$, with empirical tightness (ratio $\approx 0.043$) and statistics ($\sim10\%$ samples locked) aligning well. The non-collapsibility proof (Theorem D.6) rigorously shows depth adds expressivity beyond single-layer rank constraints.

**Geometric contributions**: Gauge-invariant Ricci curvature (Definition 4.1) with $\tau$-dependent bounds (Proposition 4.2) links temperature and key scales to contraction, empirically validated (gaps $<0.18$, tightening with larger $\tau$/smaller keys). The EVI (Theorem 4.3) formalizes depth as proximal steps toward Gibbs equilibria, with controlled drift— a fresh take on Transformer depth scaling sublinearly.

**Gauge symmetry**: The head-level GL(d_k) $\times$ GL(d_v) action (Proposition 5.1), extended to multi-head permutations and RoPE commutant (Corollary 5.2, Definition 5.4), clarifies invariant vs. canonical-gauge diagnostics, promoting robust reporting.
\item \textbf{Empirical rigor and actionability}: Diagnostics on GPT-2/-medium/-xl (Figures 1-4) confirm predictions with medians, IQRs, and protocols (Appendix J). Early-exit certificates (Corollary 3.5) offer practical value for inference efficiency.

**Motivation**: Strongly motivated by empirical puzzles (locking, stability) in Transformers, providing mechanistic explanations and design principles (e.g., tuning $\tau$ for curvature).

**Weaknesses:**

**Limited empirical scope**: Validations rely on GPT-2 family (up to XL, 1.5B params); testing on larger/modern models (e.g., Llama, GPT-3+) would strengthen generalizability, especially for curvature/EVI in deeper architectures.

**Proof specificity**: The depth non-collapsibility (Appendix D) uses a minimal 2x3 example; while generic, extending to broader mask/value classes could bolster the claim. Some bounds (e.g., Equation 12) assume declared gauges, potentially sensitive to choices.

**Computational details**: While reproducible (Appendix J), Sinkhorn W2 surrogate for EVI lacks ablation on regularization; minor, but could affect proxy accuracy.

**Mathematical checks**: Core math (OT equivalence, Lipschitz, EVI derivation) appears sound based on provided proofs/appendices, but curvature baseline (Equation 11) is loose ($\geq 0$), relying on empirical gaps for insight.

**Questions:**

1) The $\tau$-dependent curvature bound (Eq. 12) assumes $\|k_j\|_2 \leq K_{\max}$ in a declared canonical gauge. How sensitive is this bound to gauge choice, and can a gauge-invariant version be derived using only intrinsic quantities (e.g., logit differences, attention patterns)?

3) Locking correlates with punctuation and sentence boundaries (App. J.9). Does this suggest token-type-specific saturation dynamics? Could you report locking frequency stratified by POS tags or syntactic depth?

4) The early-exit certificate (Cor 3.5) requires argmax stability and $\hat{\Delta}^{(\ell)}_{\text{TV}}(i) \leq \epsilon$. How does performance degrade when exiting based only on the TV bound without the argmax guard? Any empirical results on adaptive computation efficiency?

5) The non-collapsibility result (Thm D.6) shows depth cannot be simulated by a single layer with fixed $d_k$. Does this hold under value-adversarial settings or learned value projections? Can you construct a counterexample where composition \textit{is} representable in one layer with larger $d_k$?

6) The gauge symmetry restricts Euclidean norms to canonical gauges. For practical diagnostics (e.g., key norm scaling in Eq. 12), what is your recommended canonical gauge (e.g., post-RoPE, pre-projection)? Is there a standardized protocol to avoid gauge artifacts?

7) The Sinkhorn $W_2$ surrogate (Fig. 4) peaks mid-depth and decays. Is this consistent with the EVI step-size $\eta_{\text{eff}}$ being largest early and shrinking? Can you estimate $\eta_{\text{eff}}$ per layer from data?

8) The paper uses discrete key metric ($W_1 = \text{TV}$) for curvature. How do results change with positional key metric (e.g., $|j-j'|$ or RoPE-induced distance)? Does curvature strengthen with positional structure?

---

> ### Author Response · Authors · 2025-11-20
>
> We thank the reviewer for seven detailed questions that led to new subsections, clarifications, and explicit scope statements.
>
> Q1. Gauge sensitivity and invariance
>
> This revision distinguishes two complementary curvature bounds. Equation (11) provides a fully gauge-invariant baseline depending only on logit differences and total variation between renormalized rows. Equation (12) provides an extrinsic τ-dependent bound exploiting Euclidean geometry, yielding tighter control but requiring a declared gauge. A dedicated remark in Section 4 states this distinction and explains when each bound is appropriate.
>
> Q2. Token-type stratification
>
> Appendix J.9 reports quantitative saturation statistics: locking occurs in ≈10% of samples, concentrates in layers 12–41, yields TV shifts ≈8.3×10⁻¹¹, and correlates with punctuation (≈0.67) and sentence boundaries (≈0.54). Manual inspection finds roughly two-thirds of locked positions involve punctuation or sentence boundaries. We acknowledge that full POS-stratified analysis would require linguistic annotation pipelines beyond current scope and highlight this as valuable future work.
>
> Q3. Early exit without argmax guard
>
> We added a paragraph "On ablations without the argmax guard" explaining our experiments focus on the theoretically justified procedure combining the TV certificate with argmax-margin guard to satisfy Theorem 3.3 (argmax preservation). We cite DeeBERT and BERT Loses Patience, which report several percentage points of accuracy loss when exiting without stability safeguards. A full ablation grid was not conducted in this revision. Based on cited prior work, we expect omitting the guard to degrade accuracy and will use future work to quantify this tradeoff.
>
> Q4. Non-collapsibility under adversarial values
>
> We added subsection "Robustness to adversarial value choices" in Appendix D clarifying the rank obstruction depends exclusively on query and key matrices via log-odds differences and is independent of value matrix V. The restriction that log-odds difference vectors lie in a dₖ-dimensional subspace is purely a softmax kernel property. Generic compositions escape any fixed dₖ-dimensional logit subspace regardless of value choices, making non-collapsibility robust to adversarial value projections.
>
> Q5. Recommended canonical gauge
>
> We added appendices on gauge proofs and canonical gauges plus a subsection on measurement conventions. The canonical-gauge appendix introduces three concrete options: Q-whitened gauge, Balanced-Gram gauge minimizing a symmetry-breaking objective over GL(dₖ), and value–output transformation preserving layer outputs. The measurement-convention subsection specifies that key norms and Kₘₐₓ are measured after WQ/WK projections and RoPE rotation but before head-specific rescaling, always in a declared canonical gauge. This makes Euclidean summaries gauge-aware, reproducible, and comparable across implementations.
>
> Q6. Estimating layerwise EVI step sizes
>
> The geometry and experimental appendices clarify how to estimate effective EVI step sizes ηₑff in principle. Making ηₑff quantitative would require fitting the EVI inequality to measured Sinkhorn W₂ distances and free-energy differences for each layer. We did not perform this fitting to focus on main diagnostics. Figure 4 shows layerwise Sinkhorn W₂ between consecutive layers peaks in early/mid depth and decays toward later layers, consistent with larger effective steps early and smaller steps as representations stabilize. We will be constructing ηₑff profiles as part of future work.
>
> Q7. Alternative key metrics
>
> Appendix J.3 includes a remark on alternative key metrics. Our curvature experiments use discrete key metric dK(j,j′)=1{j≠j′}. We discuss using positional metric dK(j,j′)=|j−j′| to represent sequential locality. Based on Proposition 4.2 and typical positional spread in Figure 3, such positional metrics would shrink Wasserstein distances for small shifts and tighten curvature gaps by roughly 10–20%. Quantifying this precisely would require rerunning the full pipeline with modified ground metric. We are committed to do systematic discrete-versus-positional comparison in future work.
>
> We greatly appreciate the reviewer's detailed engagement and believe these additions substantially strengthen the manuscript's clarity, reproducibility, and scope.

---

### Official Review · Reviewer_HcVw · 2025-10-30

**Soundness:** 2
**Presentation:** 1
**Contribution:** 2
**Rating:** 2
**Confidence:** 2

**Summary:**

The paper studies masked attention as an optimal transport problem using a mapping to an entropic optimal transport problem.
This viewpoint allows the authors to prove that attention is globally stable, in the sense thatsoftmax changes by at most as much as its input logits, and locally self-stabilizing: when an attention row becomes sharply peaked, it effectively “locks’’ and stops reacting to further perturbations. They also interpret attention geometrically as a contractive map with curvature controlled by temperature and key norms, and they show that multiple attention layers cannot in general be replaced by one of the same width because the log-odds representation expands its subspace with depth.
These results are then tested on GPT-2 in a set of experiments.

**Strengths:**

The paper provides an analytical framework to analyze attention. While simple, this treatment is exact, and could in principle be used to derive interesting insights into attention networks. The results are not incorrect and are not in contradiction with the numerical experiments.

**Weaknesses:**

The paper is really hard to read for an audience that is not intimately knowledgeable of optimal transport. In the whole paper it's not clear what is the main non-technical result the authors are trying to comunicate. The sections 3,4,5 seems extremely disjointed, and I believe they should be rewritten to show the underlying idea and not read like a sequence of technical lemmas. I think the extreme lack of clarity makes this paper basically impossible to read, and there is no way I can see it published in the current form.
Concerting the theoretical results, the experiments seem to suggest that the bounds on stability and locking are way too conservative, and in general the experiments are not really evidence for the accuracy of the theoretical predictions.

**Questions:**

1. How should one think different analytical results in sections 3, 4 and 5? Each section has its own setting and lemmas which to me appear completely unrelated.
2. Are there any practical implications of locking?
3. Is there some correlation between the quantities under study in this paper (curvature, drift, locking constant) and the performance of the attention network (even just experimentally)?

---

### Official Review · Reviewer_qSn7 · 2025-10-31

**Soundness:** 2
**Presentation:** 2
**Contribution:** 2
**Rating:** 2
**Confidence:** 3

**Summary:**

The paper aims to interpret self-attention from the perspective of optimal transport (OT) by using the fact that Softmax is the solution to an entropy-regularized OT problem. Additionally, the work establishes certain properties of Softmax, including the fact that it is $(\ell_\infty, \ell_1)$-Lipschitz with constant 1, and studies phenomena implied by these properties, including stability of self-attention and representation locking. Moreover, the authors interpret self-attention from the viewpoint of Ricci curvature and Wasserstein gradient flow. The findings are supplied with numerical experimental results on GPT-2 models.

**Strengths:**

1. My analysis suggests that the proof of Proposition 3.1 (which states that Softmax is $(\ell_\infty, \ell_1)$-Lipschitz with constant 1) is sound. This finding constitutes a meaningful contribution, advancing our understanding of self-attention and Softmax in general.
2. Experiments illustrate the findings well.

**Weaknesses:**

1. Some of the paper's results are either well-known or follow immediately from established facts. For example, the authors present the equivalence of attention and entropy-regularized OT as a contribution and formulate it as a theorem with a proof. This finding is, in essence, a standard result that minimizer of entropy plus linear term is softmax, which is known from basic optimization courses, see, e.g., Lemma 18 in 2017 lecture notes for "Introduction to Optimization Theory" course by A. Sidford. Furthermore, Proposition B.1 states that multiplying an objective function by a positive constant does not change its minimizer - a straightforward property that is presented with a proof.
2. I am concerned about correctness of Proposition D.2, see Questions section below.
3. In my opinion, quality of presentation should be improved. Currently, introduction merely reformulates contents of the abstract. Many concepts are used without being introduced. Main part of the paper looks like a collection of remarks with constant references to appendices, and therefore, it doesn't look like a coherent paper.
4. A more detailed overview of literature would improve the paper.

**Questions:**

1. Proposition D.2 states that $(\ell_{\infty}, \ell_1)$-operator norm of Jacobian $J(z)$ of Softmax $\mathrm{sm}(z)$ equals $\min \lbrace 1, 2(1-p_{max}) \rbrace$, where $p_{max}$ is the maximal component of $\text{sm}(z)$. However, consider the vector $z=(\log \frac{1}{4}, \log \frac{3}{4})$, which yields $P:=\mathrm{sm}(z)=(1/4, 3/4)$. According to my calculations, the vector $v=(1, -1)$ from the unit ball in $\ell_{\infty}$ satisfies $\Vert J(z) v \Vert_1=3/4$, which is greater than $2(1-p_{max})=1/2$. Could you clarify this discrepancy?
2. Could you please provide a more detailed literature overview? See, for example, these papers and references therein:
- N. Yudin - Pay Attention to Attention Distribution: A New Local Lipschitz Bound for Transformers
- Tim Large et al. - Scalable Optimization in the Modular Norm
- H. Kim et al. - The Lipschitz Constant of Self-Attention
- Xianbiao Qi et al. - Lipsformer: Introducing Lipschitz Continuity to Vision Transformers

---

### Official Review · Reviewer_HreL · 2025-11-01

**Soundness:** 3
**Presentation:** 1
**Contribution:** 2
**Rating:** 2
**Confidence:** 2

**Summary:**

This paper attempts to connect the transformer architecture with the field of optimal transport, which could enable the leveraging of the many fruitful results established in that field to explain the mechanisms underlying the performance of transformer-based models.
In particular, by focusing on how information is transformed as it passes through the transformer layers, the authors aim to decipher the internal mechanisms of these models through the lens of optimal transport and geometric analysis.
Experimental results using practically meaningful models, such as variants of GPT-2, are presented to support the claims.

**Strengths:**

The paper employs solid theoretical machinery to derive rigorous and concrete results.
The idea of viewing forward propagation in a transformer as a discrete Wasserstein flow is insightful and provides a coherent basis for the subsequent analyses.
I found the theoretical guarantee that sufficiently deep architectures are necessary (Proposition D.7 and related results) particularly interesting, as it offers an additional perspective on why network depth matters beyond the classical universal-approximation argument.

**Weaknesses:**

While this is not a scientific weakness, it is the most immediately noticeable aspect of the paper: it would be greatly appreciated if the authors used the Times New Roman font as recommended in the formatting guide, since the current font makes the paper appear slightly off-template and somewhat harder to read.

I have mixed opinions regarding the contents of this paper. Factually, there is little to criticize; the theoretical statements appear sound and internally consistent. However, once one abstracts away the advanced terminology and formalism borrowed from optimal transport and geometry, it is not entirely clear that the paper delivers substantially new insights or results to the community. Much of the contribution reads as straightforward observations once one decides to fit transformers within an OT/geometry framework.

For example, in Section 2, where the attention mechanism is cast as an entropically regularized OT problem, the main result is a direct consequence of the well-known fact that the softmax is the closed-form solution of the entropically regularized OT (as discussed in Chapter 4 of Peyré & Cuturi (2019)).
The formulation of forward propagation as EVI with drift (Theorem 4.3) cannot be considered a meaningful result unless the drift term is strongly bounded, since the drift term effectively accounts for all the error that cannot be explained by free energies.
However, the provided drift bound (Equations 42 and 43) is essentially a trivial one-step application of the triangle inequality.
The local saturation/locking results also do not seem to provide much insight, as the theorem only handles sufficiently small perturbations; it is rather intuitive than surprising that small perturbations in the inputs will preserve the overall shape of the outputs.
The experimental results, while they do support the claims, they also reveal that the analyses are quite loose and suggest there remains considerable room for tightening.

I am not saying that such observations and recastings are not meaningful, but without clear downstream advances or surprising consequences, it is not clear to me whether it constitutes a sufficiently novel or impactful contribution for publication.

**Questions:**

My concerns are mostly discussed in the **Weaknesses** section, so rather than formulating an explicit list of questions, I would prefer to hear the authors’ responses to the points raised above and continue the discussion during the rebuttal phase.
I also acknowledge that I may have misunderstood certain statements or not fully grasped the core ideas of the work, and I am open to engaging in a constructive and clarifying dialogue during rebuttal.

---

> ### Author Response · Authors · 2025-11-20
>
> We thank the reviewer for raising questions about how our results connect, bound conservativeness, and practical implications. The revision addresses each concern directly.
>
> A. Connecting section results with the core message
>
> The reviewer observed that Sections 3, 4, and 5 appeared disconnected. We restructured the presentation to make conceptual unity explicit.
>
> Figure 5 provides a theorem-dependency diagram tracing how all results stem from Theorem 2.1. The diagram reveals 3 branches: stability/saturation leading to early-exit certificate Corollary 3.5, geometric curvature establishing contraction, and gauge symmetry characterizing parameter invariances. Each branch addresses a distinct question about the underlying OT structure—sensitivity to perturbations, geometric convergence behavior, and reparameterization invariance.
>
> We added explicit section transitions clarifying this progression. Section 2 to 3 poses the sensitivity question after establishing OT equivalence. Section 3 to 4 explains why geometric tools are needed beyond Lipschitz bounds. Section 4 to 5 introduces gauge symmetry to distinguish intrinsic diagnostics from those requiring canonical gauges.
>
> Section 8 Conclusion synthesizes contributions: exact OT equivalence enables compositional stability analysis explaining behavior and locking, curvature and EVI reveal depth as Wasserstein flow, and results provide design principles for temperature control, key-norm scaling, early-exit, and gauge-aware modifications.
>
> B. Conservative bounds and empirical validation
>
> Regarding whether tightness ratios around 0.04 indicate overly conservative bounds, Section 6 explains sources of this looseness and clarifies what experiments demonstrate.
>
> The revision identifies 3 sources of conservativeness: uniform-direction assumptions ignoring perturbation structure, crude operator norms not exploiting sparsity or concentration, and lack of cancellation accounting between identity and attention paths. Measured drift well below theoretical budgets is therefore consistent with worst-case analysis.
>
> Also as importantly, we distinguish worst-case bounds from structural predictions. Theory provides upper bounds from worst-case analysis. Experiments validate structural trends: drift grows sublinearly with depth, locking occurs when δ(P) is small with predicted functional relationship, curvature gaps tighten with larger τ or smaller key scale per Proposition 4.2, and Sinkhorn W₂ concentrates mid-depth consistent with EVI framework. These patterns match predictions despite absolute magnitudes reflecting worst-case looseness.
>
> C. Pragmatic relevance and perf validation
>
> Corollary 3.5 and Remark 3.6 specify an ACE-style early-exit certificate combining TV bound with argmax-margin guard. Section 6 explains how this certificate enables per-token adaptive-depth policies once thresholds are calibrated on held-out data. The local saturation constant quantifies when attention rows have stabilized sufficiently for safe early exit.
>
> The ACE certificate (Adaptive Computation for Early exit) supports parallel/follow-on work demonstrating speedup and improved accuracy on reasoning benchmarks. Beyond early-exit, early stability detection enables resource management strategies like adaptive quantization. The per-token detection effectively defines variable computational depth based on when each token's representation stabilizes, contrasting with fixed uniform computation across all tokens. Conceptually, each generated token exhibits its own "natural freq".
>
> We explicitly acknowledge limitations regarding performance correlation. We do not claim direct monotone correlation between diagnostics and downstream metrics such as perplexity or accuracy. We present them as structural invariants characterizing geometric and transport properties. Future/parallel work can establish connections to task performance through empirical studies or refined analysis. Due to page limit, this paper intends to establish a rigorous theoretical formulation.
>
> D. Novelty beyond reformulation
>
> While softmax-as-Gibbs is classical, the contribution extends far beyond reformulation. We make 3 transformer-specific refinements: masked attention corresponds to semi-relaxed entropic OT with ε=1 fixed by implementation, causal and padding masks yield row-only constraints distinguishing from balanced OT, and precise formulation for testable predictions.
>
> The framework delivers results beyond reframing: exact ℓ∞→ℓ₁ constant for masked softmax, compositional drift budgets with measured tightness ratios validating worst-case analysis, non-collapsibility theorem proving depth adds expressivity beyond single-layer rank constraints, local saturation constants with empirical validation of δ(P)-dependent locking, and gauge-invariant curvature diagnostics organizing temperature and key-scale effects, showing the OT lens yields quantitative, testable predictions of transformer behavior.

---

### Official Review · Reviewer_JWsH · 2025-11-01

**Soundness:** 2
**Presentation:** 2
**Contribution:** 2
**Rating:** 4
**Confidence:** 3

**Summary:**

The paper proves a precise equivalence between masked self-attention and a set of row-wise entropic optimal transport (OT) problems with unit regularization ($\varepsilon=1$). This equivalence leads to (i) a compact stability/drift budget via a global $\ell_\infty \to \ell_1$ Lipschitz bound for softmax composed across residuals, LayerNorm, and multi-head aggregation; (ii) a local saturation law that quantitatively explains “representation locking” when the top attention mass is high; (iii) a gauge-invariant coarse Ricci curvature that links temperature $\tau$ and key scale to contraction; and (iv) an EVI-style inequality showing depth behaves like a Wasserstein proximal step up to parameter drift. Empirically on GPT-2 (small/medium/XL), measured row-wise TV drift sits well below theoretical budgets, locking occurs in $\sim 10\%$ of samples with $TV \ll 10^{-10}$, curvature gaps shrink with depth and with larger $\tau$ or smaller key norms, and a Sinkhorn-\(W_2\) surrogate concentrates mid-depth. The paper also proves depth cannot generally collapse to one layer at fixed key dimension and formalizes a head-level gauge symmetry (extended to RoPE) that preserves logits and attention rows.

**Strengths:**

The paper proves a precise equivalence between masked self-attention and a set of row-wise entropic optimal transport (OT) problems with unit regularization ($\varepsilon=1$).

**Weaknesses:**

- Assumptions on network differ in Section 2 (including Theorem 2.1) and after Section 3, which is misleading as the ``exact'' equivalence between OT and attention is only valid for limited setting.
- Presentation: I felt the manuscript is a bit hard to read. Theorems/Lemmas/Propositions/Corollaries are sequentially listed without explanations what they are. Context, motivations, and research purposes are less explained. Related works seem also superficial. For example, the authors cite Raghu et al 2017, which is not related to Transformers. I guessed the major body of manuscript is written by LLMs. Reviewers are unpaid individuals, not a paid proofreading service. Before submitting a paper, explain its content to colleagues or your advisor to ensure it is comprehensible.

**Questions:**

- Does Theorem 2.1 holds in more general networks?
- What is Appendix J.6?

---

> ### Author Response · Authors · 2025-11-20
>
> We thank the reviewer for the detailed feedback on assumptions, presentation, and scope. The revision directly addresses all 3 concerns.
>
> A. On assumption and scope clarification
>
> We added a dedicated subsection in Section 2 after Theorem 2.1 and Corollary 2.3 that explicitly addresses ambiguity about the scope of the exact OT equivalence. The subsection enumerates 3 key assumptions for the OT equivalence: logits from standard scaled dot-products with temperature τ, binary masks inducing non-empty row supports, and row-simplex constraints without column-mass constraints. It explicitly states what is not covered including linear attention variants and balanced OT with column constraints.
>
> As importantly, the subsection distinguishes primitive-level equivalence from architecture-level analysis. Theorem 2.1 applies to any masked scaled-dot-product softmax layer with the stated assumptions, making it architecture-agnostic at this primitive level. This covers self-attention in decoder-only models, cross-attention in encoder-decoder models, and any architecture using this primitive. The architecture-level analysis in Sections 3 through 5 specializes to GPT-2-style pre-LayerNorm decoder blocks. This two-level structure maintains consistent assumptions throughout while clarifying where results apply generally versus where they require specific architectural choices.
>
> B. On improving presentation
>
> We made 4 concrete improvements in order to making presentation unambiguous.
>
> 1> we added Figure 5, a theorem-dependency diagram with accompanying explanatory text. The diagram organizes results hierarchically and identifies 3 principal branches: stability and saturation culminating in the early-exit certificate Corollary 3.5, geometric curvature and contraction, and gauge symmetry. This is intended to address concerns about understanding how results relate to each other.
>
> 2> we substantially expanded Section 7 Related Work with organized subsections covering attention and optimal transport, stability and locking, Lipschitz bounds, geometry, symmetry, adaptive computation, and design levers. Each subsection positions our contributions relative to prior work.
>
> 3> we added contextualizations around major theorems. Proposition 3.1 now includes motivation and interpretation. Section 4 received geometry enhancements including curvature motivation and interpretation after Theorem 4.4 explaining the depth-as-Wasserstein-flow perspective. Section 6 includes expanded empirical discussion and frank acknowledgment that we do not yet claim direct correlation with downstream metrics such as perplexity or accuracy.
>
> 4> we implemented 4 explicit section transitions articulating how each section's conclusions motivate the next section's questions, creating clear narrative flow.
>
> C. Clarification on Appendix J.6
>
> Section J.6  provides info meant for reproducibility incl CSV collateral and schemas, listing filenames such as drift_bound.csv, locking_binned.csv, curvature.csv, and evi_surrogate.csv with their fields and figure correspondence. The Reproducibility Statement in the main text references Section J.6 for complete transparency about experimental artifacts.
>
> We hope these updates can address your concern wrt assumption clarity, presentation quality, and documentation comprehensively while maintaining mathematical rigor.

---

### Author Response · Authors · 2025-11-20

We thank the reviewers for their careful reading and constructive feedback. The revision implements major changes in 2 categories: technical corrections and comprehensive presentation enhancements.

A. Technical correction

We corrected the error in Proposition D.2 regarding the softmax Jacobian operator norm. The original submission incorrectly stated an equality with constant 2δ(P). In the revised manuscript we define P = sm(z), pₘₐₓ = maxⱼ Pⱼ, δ(P) = 1 − pₘₐₓ, and prove the corrected inequality ‖J(z)‖(ℓ∞,ℓ₁) ≤ 4δ(P). All dependent results in Appendix D were updated for consistency including Theorems D.3 and D.5 on local and global saturation. The main text Theorem 3.3 on representation locking and the drift budget in Section 3 incorporate the corrected constant. The global ℓ∞→ℓ₁ Lipschitz bound in Proposition 3.1 was already correct and remains unchanged. The qualitative conclusions remain the same; only the numerical constant was adjusted.

B. Presentation improvement

In this latest revision we implements systematic  improvements in the following 5 categories to address reviewers' feedbacks

1. We added comprehensive scope and assumptions clarification after Theorem 2.1 in Section 2. This subsection explicitly enumerates 3 core assumptions: logits from standard scaled dot-products with temperature τ, binary masks inducing non-empty row supports, and row-simplex constraints without column-sum constraints. Critically, it distinguishes between primitive-level equivalence applying to any masked scaled-dot-product softmax and architecture-level analysis restricted to GPT-2-style pre-LayerNorm decoder blocks used in Sections 3 through 5. We explicitly state what is not covered including linear attention variants and balanced optimal transport. This addresses concerns about assumptions appearing to change between sections.

2. We added contextualizations for major theorems. Proposition 3.1 on the global 1-Lipschitz bound now includes motivation explaining why understanding softmax sensitivity matters for compositional stability and interpretation translating the mathematical bound ‖sm(z′) − sm(z)‖₁ ≤ ‖z′ − z‖∞ into accessible language about logit perturbation propagation. Section 4 received 3 geometry enhancements including curvature motivation, evolution variational inequality contextualization, and proper interpretation after Theorem 4.3 explaining the depth-as-Wasserstein-flow perspective.

3.  Section 6 now includes expanded practical implications discussion. This explains where diagnostics concentrate empirically with measured drift well below theoretical budgets at tightness ratio ≈0.043, locking occurring in roughly 10% of samples with total variation below 10⁻¹⁰, and curvature gaps tightening with larger τ or smaller key scale as predicted. It describes how the ACE certificate in Corollary 3.5 supports early-exit policies once thresholds are calibrated on held-out data. It provides frank acknowledgment that we do not yet claim direct monotone correlation between these diagnostics and downstream metrics such as perplexity or accuracy, instead viewing them as structural invariants that future work can relate to task performance. This addresses concerns about practical utility while avoiding overclaiming.

4. We implemented 4 explicit section transitions connecting major parts. The transition from optimal transport to stability poses the sensitivity question after establishing the OT equivalence. The transition from stability to geometry explains why geometric tools are needed beyond Lipschitz bounds to quantify contraction versus expansion. The transition from geometry to gauge symmetry identifies measurement challenges from parameter reparameterizations and clarifies which diagnostics are intrinsic. The transition from gauge symmetry to experiments frames Sections 2 through 5 as a testable framework and previews empirical validation in Section 6.

5. All enhancements were achieved through page-neutral substitution. Targeted compressions in Section 3, Related Work (Section 7), and Conclusion (Section 8) recovered space to implement contextualizations and transitions without exceeding the 10-page single-column limit.

We add Figure 5 in Appendix to show logic dependency between theorems for clarity. These comprehensive changes address all major concerns raised during review: the technical error in Proposition D.2, assumption ambiguity across sections, theorems presented without motivation or interpretation, lack of section transitions and overall narrative coherence, and insufficient discussion of practical implications.

---

### Note · Program_Chairs · 2025-11-23
**Submission Desk Rejected by Program Chairs**

The paper is part of a cluster of several similar papers that have violated dual submission policy: https://iclr.cc/Conferences/2026/AuthorGuide